# No Double Descent in PCA: Training and Pre-Training in High Dimensions

## Abstract

With the recent body of work on overparameterized models the gap between theory and practice in contemporary machine learning is shrinking. While many of the present state-of-the-art models have an encoder-decoder architecture, there is little theoretical work for this model structure. To improve our understanding in this direction, we consider linear encoder-decoder models, specifically PCA with linear regression on data from a low-dimensional manifold. We present an analysis for fundamental guarantees of the risk and asymptotic results for isotropic data when the model is trained in a supervised manner. The results are also verified in simulations and compared with experiments from real-world genetics data. Furthermore, we extend our analysis to the popular setting where parts of the model are pre-trained in an unsupervised manner by pre-training the PCA encoder with subsequent supervised training of the linear regression. We show that the overall risk depends on the estimates of the eigenvectors in the encoder and present a sample complexity requirement through a concentration bound. The results highlight that using more pre-training data decreases the overall risk only if it improves the eigenvector estimates. Therefore, we stress that the eigenvalue distribution determines whether more pre-training data is useful or not.

## 1 Introduction

Many recent success stories of deep learning employ an encoder-decoder structure, where parts of the model are pre-trained in an unsupervised or self-supervised way. Examples can be found in computer vision (Caron et al., 2020; Chen et al., 2020; Goyal et al., 2021), natural language processing (Vaswani et al., 2017; Devlin et al., 2019; Raffel et al., 2020) or multi-modal models (Ramesh et al., 2021; Alayrac et al., 2022). Understanding the properties of this model structure might shed light on how to reliably build large-scale models.

We add to the theoretical understanding of encoder-decoder based models by studying a model consisting of PCA and a linear regression head. We analyse this model for the supervised case and for the case where unsupervised pre-training is followed by supervised linear regression. Our model can be viewed as a simplified, linear example of a large pre-trained deep neural network in combination with linear probing (Devlin et al., 2019; Schneider et al., 2019). While linear models do not reveal the whole picture, they are studied as a tractable, first step towards deeper understanding. Indeed, research on linear models has previously provided important insights into relevant mechanisms (Saxe et al., 2014; Lampinen & Ganguli, 2019; Arora et al., 2019; Gidel et al., 2019; Pesme et al., 2021).

We utilize data generated from a low-dimensional manifold, similar to Goldt et al. (2020). This is motivated by the manifold hypothesis (Fefferman et al., 2016) which states that real-world high-dimensional data often have an underlying low-dimensional representation. Our PCA encoder can exploit this data structure effectively. While we keep the low-dimensional data structure fixed, we vary the number of features w.r.t. the number of training data points which allows us to analyse what is often referred to as overparameterization, i.e. data features or model parameters than training samples (Belkin et al., 2019). We do not consider parameter count, since for our model the number of parameters, i.e. the linear regressors, stay fixed due to the PCA encoding. Instead we analyse high-dimensional settings. Studying overparameterization gives theoretical justification of the success of modern large-scale neural networks such as Szegedy et al. (2016); Dosovitskiy et al. (2021).

Theoretical grounding is exceeded by the empirical success of machine learning and specifically deep learning methods through new model structures (Krizhevsky et al., 2017; He et al., 2016; Vaswani et al., 2017) or training methods (Erhan et al., 2010; Ioffe & Szegedy, 2015; Ba et al., 2016). In recent years our theoretical understanding grew e.g. through the analysis of implicit regularization (Gunasekar et al., 2017; Chizat & Bach, 2020; Smith et al., 2021). But also experimental work contributed to our understanding (Keskar et al., 2017; Zhang et al., 2017). The goal of this paper is to extend our understanding of the successful encoder-decoder model structure through theoretical analysis of PCA-regression and by extensive numerical simulations. We generalize results for linear regression and combine classical analysis of overparamterization with pre-training of model components. Our contributions can be summarized as:

- In the **supervised case**, we provide **theoretical guarantees** for the risk and parameter norm of the PCA-regression model. For isotropic data we extend the results to the limit where the number of data points $n$ and features $m$ tend to infinity such that $m/n \to \gamma$.

- Through **simulations**, we confirm our theory for isotropic data and explain the model behavior on data from a low-dimensional manifold. Using **genetics data**, we validate our findings in a high-dimensional real-world example.

- We extend our analysis to the popular scenario of unsupervised **pre-training of the encoder** and show that the correct estimation of feature covariance eigenvectors is crucial for low risk. These estimates are highly dependent on the data structure through the **eigenvalue decay rate**. The results provide a link to known asymptotic results by Xu & Hsu (2019). We challenge the common wisdom that more pre-training data improves the overall risk and show that this is the case only if it improves the estimate of the eigenvectors in the encoder which is e.g. the case in data with rapidly decaying eigenvalues.

## 2 RELATED WORK

**Overparameterization**  The study of overparameterized models offers a natural route to gain theoretical understanding when it comes to the successes of large models with good generalization properties (Neyshabur et al., 2015; Zhang et al., 2017). The double descent was discovered and analysed in early works (Krogh & Hertz, 1991; Geman et al., 1992; Opper, 1995) but the framing as 'double descent' (Belkin et al., 2019) boosted research in this direction even if generalization of large models was already studied before (Bartlett & Mendelson, 2002; Dziugaite & Roy, 2017; Belkin et al., 2018; Advani et al., 2020). We add to the understanding of machine learning models by analysing the neglected class of encoder-decoder models with the PCA-regression model.

**Analysis of pre-training**  The introduction of pre-training of neural networks was a paradigm shift for deep learning. Empirical work (Erhan et al., 2010; Raghu et al., 2019) but also theoretical work such as for sample complexity (Tripuraneni et al., 2020; Du et al., 2021) or the out-of-distribution risk (Kumar et al., 2022) tried to understand the mechanisms. For unsupervised pre-training, contrastive methods were studied (Wang & Isola, 2020; Von Kügelgen et al., 2021). Encoder-decoder based autoencoders are analysed for training dynamics (Nguyen et al., 2019; 2021) or overparameterization (Radhakrishnan et al., 2019; 2020; Buhai et al., 2020; Zhang et al., 2020). In contrast, we study pre-trained PCA encoders and relate the risk to the covariance estimation of the encoder.

**Latent variable data generator**  We generate data via a linear latent variable data generator based on a low-dimensional manifold. The hidden manifold model (Goldt et al., 2020) and random feature model (Rahimi & Recht, 2007) present similar but nonlinear models. Goldt et al. (2022); Hu & Lu (2022) showed that these nonlinear models are asymptotically equivalent to linear Gaussian models under assumptions such as that the latent dimension $d \to \infty$. In contrast, we keep this dimension fixed. Asymptotic generalization results for this data generator are presented in Gerace et al. (2020); Mei & Montanari (2022). Different to our work where we exploit the low-dimensional structure with the PCA-regression model, they do not use this information by using Ridge or logistic regression.

**PCA-regression model**  Using PCA (Jolliffe, 1982) is common—discussions focus on the choice of principle components (Breiman & Freedman, 1983) or its use for high-dimensional data (Lee et al., 2012). PCA-regression is investigated in Xu & Hsu (2019) for general but fully known covariances in the asymptotic regime. Wu & Xu (2020) extend it by showing that the misalignment of true

and estimated eigenvectors affect the risk. Huang et al. (2022) use misalignment bounds (Loukas, 2017) to remove the known covariance assumption and obtain non-asymptotic risk bounds. Our work fills the gaps by providing asymptotic results for isotropic data. We generalize Loukas (2017) to obtain a sample complexity for the covariance estimation in the PCA which is the missing piece to quantify when the results from Xu & Hsu (2019) can be used in practice with pre-training. It turns out that the data covariance structure is crucial as Wainwright (2019) points out.

## 3 PROBLEM FORMULATION

**Data generator** We generate a data set $\{\boldsymbol{x}_i, y_i\}_{i=1}^n$ according to a latent variable data generator

$$\boldsymbol{x}_i = \boldsymbol{D}\boldsymbol{z}_i + \boldsymbol{e}_i, \tag{1}$$

$$y_i = \boldsymbol{\theta}^\top \boldsymbol{z}_i + \varepsilon_i, \tag{2}$$

by mapping the latent variable $\boldsymbol{z}_i \in \mathbb{R}^d$ with $\boldsymbol{D} \in \mathbb{R}^{m \times d}$ into the observed features $\boldsymbol{x}_i \in \mathbb{R}^m$ and with $\boldsymbol{\theta} \in \mathbb{R}^d$ into the observed outputs $y_i \in \mathbb{R}$. We create $\boldsymbol{D}$ randomly such that $\|\boldsymbol{D}\|_F^2 = dc^2$ with $c$ as correction factor to control the signal-to-noise ratio (SNR), defined in (27). Similarly, to control the outcome-noise-ratio we create $\boldsymbol{\theta}$ such that $\mathbb{E}\left[\|\boldsymbol{\theta}^\top \boldsymbol{z}\|_2^2\right] = r_{\boldsymbol{\theta}}^2$. Feature noise $\boldsymbol{e}_i \sim \mathcal{N}(\boldsymbol{0}, \boldsymbol{I}_m)$ and output noise $\varepsilon_i \sim \mathcal{N}(0, \sigma_y^2)$ are added. The latent variables are generated such that the singular values of the features have an exponential decay controlled by the decay rate $\alpha \geq 0$ according to

$$\boldsymbol{z}_i \sim \mathcal{N}(0, \lambda_i^2 \boldsymbol{I}_d) \quad \text{with} \quad \lambda_i^2 = \exp(-i\alpha). \tag{3}$$

Our theoretical results do not specifically require an exponential decay of the eigenvalues or a specific rate. However, fast decaying eigenvalues occur in many real-world examples, see Appendix B. We distinguish between two data generators:

1. *Isotropic data.* This is a special case of (1), (2) with $d = m$, $\boldsymbol{D} = \boldsymbol{I}_m$, $\alpha = 0$ and $\boldsymbol{e} = \boldsymbol{0}$ to generate isotropic features. It allows us to rewrite the data generator as

$$y_i = \boldsymbol{\theta}^\top \boldsymbol{x}_i + \varepsilon_i \quad \text{with} \quad \boldsymbol{x}_i \sim \mathcal{N}(0, \boldsymbol{I}_m). \tag{4}$$

2. *Latent variable data.* We distinguish between 1) $\alpha = 0$ leading to an isotropic but low-dimensional signal and 2) $\alpha > 0$ which has dominant, but rapidly decaying eigenvalues of the feature covariance matrix. The latter data generator is motivated since many real-world data sets have a low-dimensional signal manifold with rapidly decaying eigenvalues.

Note that while our latent variable data generator is similar to the latent space model from Hastie et al. (2022), we use our PCA-regression model instead of direct regression from features to outputs. A graphical model of our data generator is provided in Figure 1 and details are in Appendix C.

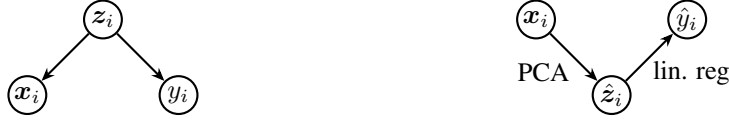

Figure 1: **Problem formulation.** *Left:* Data generator. *Right:* PCA and linear regression model.

**Model** We use a linear model which resembles an encoder-decoder based architecture. The input data $\boldsymbol{x}_i$ is encoded into a lower $\hat{d}$-dimensional space $\hat{\boldsymbol{z}}_i$ via the use of PCA, where $\hat{d}$ is chosen during model selection. When it comes to the decoder we employ a linear regression model with parameters $\hat{\boldsymbol{\theta}} \in \mathbb{R}^{\hat{d}}$. The resulting model is visualized in Figure 1 and we can formulate it as

$$\hat{\boldsymbol{z}}_i = \hat{\boldsymbol{V}}^\top \boldsymbol{x}_i, \tag{5}$$

$$\hat{y}_i = \hat{\boldsymbol{\theta}}^\top \hat{\boldsymbol{z}}_i. \tag{6}$$

Collecting the features as rows of the data matrix $\boldsymbol{X} = \begin{bmatrix} \boldsymbol{x}_1^\top & \dots & \boldsymbol{x}_n^\top \end{bmatrix}^\top$, we compute the principal components $\hat{\boldsymbol{V}} \in \mathbb{R}^{m \times \hat{d}}$ as the $\hat{d}$ first right singular vectors of the rank $\hat{d}$ reducing SVD $\boldsymbol{X} \approx \hat{\boldsymbol{U}}\hat{\boldsymbol{\Sigma}}\hat{\boldsymbol{V}}^\top$. Note that the estimated latent dimension $\hat{d}$ can be different from the true latent dimension $d$ of the latent variable data generator. We refer to this construction as the *PCA-regression* model. In our numerical results we compare with a model which directly regresses the outcomes from the features, referred to as the *direct regression* model.

## 4 ANALYZING THE SUPERVISED CASE

Training the complete PCA-regression model in a supervised way represents a situation commonly encountered in high-dimensional real-world applications. Examples of using this model are in exploratory statistical research (Massy, 1965), econometrics (Geweke, 1996), genetics (Wang & Abbott, 2008), robotics (Vijayakumar & Schaal, 2000) and many more.

### 4.1 THEORETICAL ANALYSIS

For our analysis, we are interested in closed form solutions for the risk and parameter norm in order to obtain fundamental guarantees for the PCA-regression model. We decompose our solution into bias and variance terms similar to classic decompositions and interpret the results.

**Bias-variance decomposition** We stack all outputs in the vector $\boldsymbol{y} \in \mathbb{R}^n$ and all estimated latent variables as rows in the matrix $\hat{\boldsymbol{Z}} \in \mathbb{R}^{n \times \hat{d}}$. The solution to the unregularized linear regression yields

$$\hat{\boldsymbol{\theta}} = (\hat{\boldsymbol{Z}}^\top \hat{\boldsymbol{Z}})^+ \hat{\boldsymbol{Z}}^\top \boldsymbol{y} = (\hat{\boldsymbol{V}}^\top \boldsymbol{X}^\top \boldsymbol{X} \hat{\boldsymbol{V}})^+ \hat{\boldsymbol{V}}^\top \boldsymbol{X}^\top \boldsymbol{y}, \tag{7}$$

where $(\cdot)^+$ denotes the Moore-Penrose pseudoinverse. We can rewrite our data generator directly from features to outputs as $\boldsymbol{y} = \boldsymbol{X}\boldsymbol{\beta} + \boldsymbol{\epsilon}$ with $\boldsymbol{\beta}^\top = \boldsymbol{\theta}^\top \boldsymbol{D}^+ \in \mathbb{R}^m$ and $\epsilon_i \sim \mathcal{N}(0, \sigma_\epsilon^2)$ where $\sigma_\epsilon^2 = \sigma_y^2 + ||\boldsymbol{\beta}||_2^2$. Following Appendix D.2 the solution becomes

$$\hat{\boldsymbol{\theta}} = (\hat{\boldsymbol{\Sigma}}^\top \hat{\boldsymbol{\Sigma}})^+ \hat{\boldsymbol{\Sigma}}^\top \hat{\boldsymbol{U}}^\top (\boldsymbol{X}\boldsymbol{\beta} + \boldsymbol{\epsilon}) = \hat{\boldsymbol{V}}^\top \boldsymbol{\beta} + \hat{\boldsymbol{\Sigma}}^+ \hat{\boldsymbol{U}}^\top \boldsymbol{\epsilon}. \tag{8}$$

**Lemma 1.** *Let the feature sample covariance be $\hat{\boldsymbol{C}} = \frac{1}{n} \boldsymbol{X}^\top \boldsymbol{X}$ and the true covariance be $\boldsymbol{C}$. Define the orthogonal projectors $\boldsymbol{\Phi} = \hat{\boldsymbol{V}} \hat{\boldsymbol{V}}^\top$ and $\boldsymbol{\Pi} = \boldsymbol{I}_m - \boldsymbol{\Phi}$, where $\boldsymbol{\Phi}$ is the projection onto the column space of the $\hat{d}$ first right singular vectors of $\boldsymbol{X}$. Then, the risk of the PCA-regression model $R(\hat{\boldsymbol{\theta}}) = \mathbb{E}_{(\boldsymbol{x}_0, y_0)} \left[ (y_0 - \hat{y}(\boldsymbol{x}_0)^2 \right]$ and the parameter norm $\|\hat{\boldsymbol{\theta}}\|_2^2 = \hat{\boldsymbol{\theta}}^\top \hat{\boldsymbol{\theta}}$ are given by*

$$\mathbb{E}_{\boldsymbol{\epsilon}} \left[ R(\hat{\boldsymbol{\theta}}) \right] = \boldsymbol{\beta}^\top \boldsymbol{\Pi} \boldsymbol{C} \boldsymbol{\Pi} \boldsymbol{\beta} + \frac{\sigma_\epsilon^2}{n} \operatorname{Tr}(\hat{\boldsymbol{V}}^\top \boldsymbol{C} \hat{\boldsymbol{V}} \hat{\boldsymbol{V}}^\top \hat{\boldsymbol{C}}^+ \hat{\boldsymbol{V}}) + \sigma_\epsilon^2, \tag{9}$$

$$\mathbb{E}_{\boldsymbol{\epsilon}} \left[ \|\hat{\boldsymbol{\theta}}\|_2^2 \right] = \boldsymbol{\beta}^\top \boldsymbol{\Phi} \boldsymbol{\beta} + \frac{\sigma_\epsilon^2}{n} \operatorname{Tr}(\hat{\boldsymbol{V}}^\top \hat{\boldsymbol{C}}^+ \hat{\boldsymbol{V}}). \tag{10}$$

The proofs are in Appendices D.3, D.5. In both equations, the variance (second) term is controlled by the estimated singular vectors $\hat{\boldsymbol{V}}$, which project the covariances $\boldsymbol{C}, \hat{\boldsymbol{C}}$ to a $\hat{d}$-dimensional subspace and therefore contain less noise. Hence, we expect the variance term to decrease constantly for larger $\gamma$ and that the PCA-regression model therefore avoids the "interpolation peak" at $\gamma = 1$ which linear regression has. The results generalize Lemma 1 in Hastie et al. (2022) for the risk of direct regression models since we obtain the same form when choosing $\hat{d} = m$, i.e. no dimensionality reduction.

**Asymptotics for isotropic features** Using results from random matrix theory, and Lemma 1 we derive asymptotics for the risk and parameter norm in the case of isotropic features $\boldsymbol{C} = \boldsymbol{I}_m$.

**Theorem 1.** *Assume isotropic features $\boldsymbol{C} = \boldsymbol{I}_m$, which implies $d = m$ and choose constant $\hat{d}$. Then, as $m, n \to \infty$, such that $\frac{m}{n} \to \gamma$, the expected risk and parameter norm satisfy almost surely*

$$\mathbb{E}_{\boldsymbol{\epsilon}} \left[ R(\hat{\boldsymbol{\theta}}) \right] \to \sigma_\epsilon^2 \frac{m}{n} \int_{\bar{s}}^\infty \frac{1}{s} dF_\gamma(s) + \sigma_\epsilon^2 + \begin{cases} \boldsymbol{\beta}^\top \boldsymbol{\beta} \left( 1 - \min(\hat{d}, m)/m \right) & \text{for} \quad \gamma < 1 \\ \boldsymbol{\beta}^\top \boldsymbol{\beta} \left( 1 - \min(\hat{d}, n)/m \right) & \text{for} \quad \gamma > 1 \end{cases}, \tag{11}$$

$$\mathbb{E}_{\boldsymbol{\epsilon}} \left[ \|\hat{\boldsymbol{\theta}}\|_2^2 \right] \to \sigma_\epsilon^2 \frac{m}{n} \int_{\bar{s}}^\infty \frac{1}{s} dF_\gamma(s) + \begin{cases} \boldsymbol{\beta}^\top \boldsymbol{\beta} \min(\hat{d}, m)/m & \text{for} \quad \gamma < 1 \\ \boldsymbol{\beta}^\top \boldsymbol{\beta} \min(\hat{d}, n)/m & \text{for} \quad \gamma > 1 \end{cases}, \tag{12}$$

*with $F_\gamma$ the Marčenko-Pastur law (Marčenko & Pastur, 1967) and $\bar{s}$ the value in $\mathbb{R}$ that satisfies $\frac{\hat{d}}{m} = \int_{\bar{s}}^\infty dF_\gamma$. In both equations, the first term represents the variance and the last one the bias.*

The proofs are in Appendices D.4, D.6. Again, we obtain the same risk when choosing $\hat{d} = m$ as for direct regression models on isotropic data, see Theorem 1 in Hastie et al. (2022). Contrary to direct regression, the PCA-model will always have a bias term since $\hat{d} < m, n$ in general.

## 4.2 NUMERICAL RESULTS

In this section we give numerical results for the different data generators and compare these results with those from the analysis above. We compare our PCA-regression model with 1) the learnt *direct regression* model and 2) a model that predicts always zero which we denote as *null risk*.

**Isotropic features**  We generate $n = 400$ data points for training and testing according to our isotropic data generator (4), implying $d = m$ with $\sigma_\varepsilon^2 = 1$ and $r_{\boldsymbol{\theta}}^2 = 1$. Each sample has $m = \gamma n$ features where we vary $\gamma \in [0.3, 20]$, i.e. from low-dimensional ($\gamma < 1$) to high-dimensional ($\gamma > 1$) features. We compute risk $R(\hat{\boldsymbol{\theta}})$ and parameter norm $||\hat{\boldsymbol{\theta}}||_2^2$ as in the definition of Lemma 1 and average over 200 realizations. The results are compared with analytical solutions from Theorem 1.

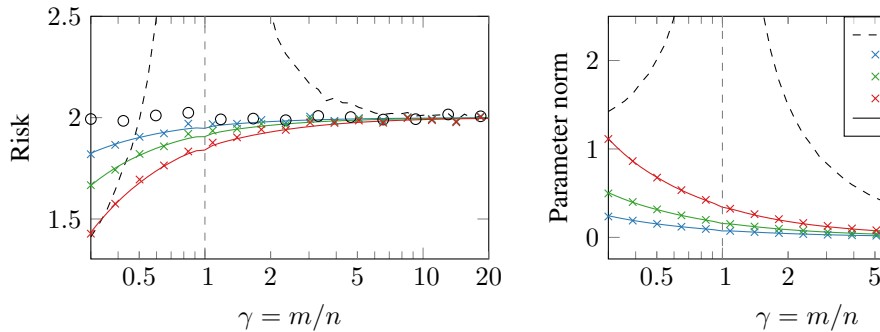

Figure 2: **Supervised results on isotropic data: analysis vs simulation.** Solid lines: analytical solutions (Theorem 1); '×': avg. simulation results; '○': null risk. *Left:* Risk. *Right:* Parameter norm.

Figure 2 depicts the results for different values of $\hat{d}$; we can make several observations: 1) The numerical results, i.e. the '×' marks, and the analytical solutions, i.e. the solid lines, align perfectly and therefore support our theoretical analysis. The expected decrease of the variance term and a nonzero bias term for all $\gamma$ can be observed in Figure 10 where we show the bias-variance decomposition according to Theorem 1; 2) For sufficiently large $\hat{d}$ the results of the PCA model match the direct regression results in the limit of small and large $\gamma$. For isotropic data, every singular direction is equally important and the PCA requires sufficiently many components, i.e. large $\hat{d}$ to achieve reasonable results; 3) The PCA-regression model does not suffer from the singularity at $\gamma = 1$ as we predicted from Lemma 1. The PCA alleviates the bad conditioning of the matrix $\boldsymbol{X}^\top \boldsymbol{X}$ which has to be inverted for the least squares solution. Below we will see that ridge regression has a similar effect; and, 4) The parameter norm is constantly decreasing for larger $\gamma$. We observe this for all models, which implies that we obtain smooth solutions which are beneficial to avoid overfitting.

**Latent variable data**  We use the latent variable data generator with $d = 20$, $r_{\boldsymbol{\theta}}^2 = 1$, $\sigma_y^2 = 0$, feature SNR $\rho_{\boldsymbol{x}} = 1$ and $\boldsymbol{\theta}$ as in (33) to generate $n = 400$ training and testing data points and average over 200 realizations. The results for the risks are depicted in Figure 3 for eigenvalue decay of $\alpha = 0$ (left) and for $\alpha = 0.25$ (right). Corresponding plots for the parameter norm are in Figure 11.

We observe for $\alpha = 0$ (left plot) if $\hat{d} \geq d$, then the PCA-regression model approaches the direct regression results for small and large $\gamma$. The plots for $\hat{d} = 20$ and $\hat{d} = 40$ overlay since both are larger than $d$ and capture all information. However, for misspecified models with $\hat{d} < d$ the solution obtained for the PCA-regression is suboptimal. Following Lemma 1, by choosing $\hat{d} < d$ we remove important eigendirections and therefore observe an increased risk. Similar conclusions can be drawn for the results for data with $\alpha > 0$ (right plot) but with less penalty on the risk for suboptimal $\hat{d}$.

**Real-world example: Genetics**  To visualize the PCA-regression model under high-dimensional inputs for a real-world data example, we use the Diverse MAGIC wheat data set (Scott et al., 2021) from the National Institute for Applied Botany. The data set contains the genome sequence of 504 inbred wheat lines and multiple phenotypes. We split the data in 252 training samples and equally many test samples. There are 1.1 million nucleotides in the genotype sequences which are

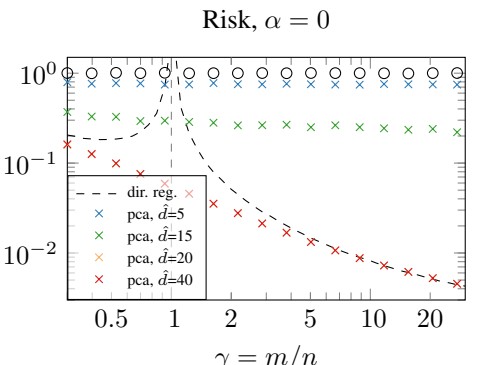 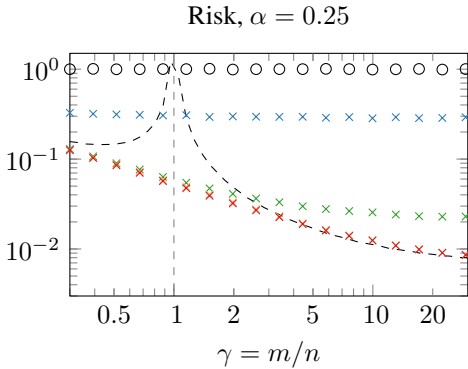

Figure 3: **Supervised risk on latent variable data: simulation**. *Left:* Risk of models for data generated with feature covariance eigenvalue decay of $\alpha = 0$. *Right:* Results with $\alpha = 0.25$.

binary encoded as difference to a reference sequence. We subsample the genotypes uniformly for a varying number of features $m$. As outcome we use one of the real-values phenotypes.

Figure 4 shows the median results over 100 realizations for different latent dimension $\hat{d}$. We observe a qualitative resemblance to the results for the latent variable model in Figure 3. 1) The PCA-regression risk decreases monotonically with increasing $\gamma$ and 2) higher values of $\hat{d}$ reach the lowest overall risk. Different is that the PCA-regression does not reach the same level as the direct regression for larger $\gamma$. However, this is reasonable since 1) the eigenvalue distribution in the genetics example is heavy tailed (see Figure 16) which implies that the true latent dimension would be much larger. Further, 2) the relationship between genotypes and phenotypes may not be linear in nature.

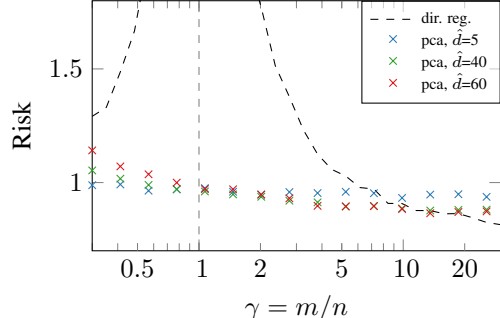

Figure 4: **Supervised risk for real-world example.** Diverse MAGIC wheat genetics data set.

## 5 PRE-TRAINING THE PCA ENCODER

So far, we analysed the case when the complete model is trained supervised. Now we extend to the popular case of pre-training parts of the model in an unsupervised way. In this context we can view our model as a simple, linear version of large pre-trained neural networks with linear probing. Our analysis therefore yields insights to their understanding. The pre-training extension requires a generalization of our theory because we deal with different data sets of varying size.

### 5.1 GENERALISATION OF PROBLEM FORMULATION

First, we pre-train the PCA on a so called pre-training data set $\{\boldsymbol{x}_i\}_{i=1}^{n_p}$ without output values $y_i$. It can therefore only be used for unsupervised pre-training. Second, we train only the linear regression head on the PCA features with the training data set $\{\boldsymbol{x}_i, y_i\}_{i=1}^{n}$. Note that the number of samples $n_p$ in the pre-training data set differs from the number of samples $n$ in the training data set.

**Data generator** In this section, we focus on the latent variable data generator. We change our feature generation from (1) to simplify the theoretical analysis. We orthogonalise the signal $\boldsymbol{z}$ (generated by (3)) and the noise $\boldsymbol{e}$ by introducing $\boldsymbol{D}_\perp$ such that $\boldsymbol{D}^\top \boldsymbol{D}_\perp = \boldsymbol{0}$. We use the following feature generator for both, the training and pre-training data set features

$$\boldsymbol{x}_i = \boldsymbol{D}\boldsymbol{z}_i + \boldsymbol{D}_\perp \boldsymbol{e}_i. \tag{13}$$

**Model** The model is the same as in the supervised case. Since the PCA is performed on the pre-training data set, we rename the first $\hat{d}$ estimated eigenvectors of the feature covariance matrix from the pre-training data set to $\hat{\boldsymbol{H}}$. We do so to distinguish it from the eigenvectors $\hat{\boldsymbol{V}}$ estimated using the training data set as in the supervised case. Hence, in the pre-training case we obtain

$$\hat{\boldsymbol{z}}_i = \hat{\boldsymbol{H}}^\top \boldsymbol{x}_i. \tag{14}$$

## 5.2 THEORETICAL ANALYSIS

As in section 4.1 we want to establish a connection to the complete training risk as fundamental model guarantee. Since we extend the setting to pre-train the encoder on a different data set, we have to deal with sample complexities for the estimation of eigenvectors in the PCA.

With the orthogonal feature generator (13), we recover the true latent variables from features

$$\boldsymbol{z}_i = \boldsymbol{D}^+(\boldsymbol{x}_i - \boldsymbol{D}_\perp \boldsymbol{e}_i) = \boldsymbol{D}^+ \boldsymbol{x}_i. \tag{15}$$

Comparing it with the projection from the model in (14), we notice that the estimated latent space depends on how well $\hat{\boldsymbol{H}}^\top$ estimates $\boldsymbol{D}^+$. Hence, the risk analysis problem in the case with pre-training turns into a sample complexity problem of the eigenvectors. Note that $\boldsymbol{D}^+ = \boldsymbol{D}^\top/c^2$ with correction factor $c$ for SNR control, see Appendix C.

**Estimation of eigenvectors** The sample complexity of eigenvectors is thoroughly studied by Loukas (2017). Here, we review some of their results and adapt them to our setting. The PCA loss of encoding $\boldsymbol{x}$ into the (estimated) latent space is given by

$$\mathcal{L}(\boldsymbol{D}) = \mathbb{E}\left[\|\boldsymbol{x}\|_2^2 - \|\boldsymbol{D}^+\boldsymbol{x}\|_2^2\right] = \sum_{i=d+1}^m s_i, \tag{16}$$

$$\mathcal{L}(\hat{\boldsymbol{H}}) = \mathbb{E}\left[\|\boldsymbol{x}\|_2^2 - \|\hat{\boldsymbol{H}}^\top\boldsymbol{x}\|_2^2\right] = \sum_{i=1}^m s_i - \sum_{i=1}^{\hat{d}}\sum_{j=1}^m (\hat{\boldsymbol{h}}_i^\top \boldsymbol{h}_j)^2 s_j. \tag{17}$$

Here, $s_i$ is the $i$th eigenvalue and $\boldsymbol{h}_i$ is the $i$th eigenvector of the true feature covariance matrix. The difference of the PCA losses, quantifies how well a sample $\boldsymbol{x}$ is projected with the estimated eigenvectors $\hat{\boldsymbol{H}}$ into the latent space compared to a projection with the true eigenvectors $(\boldsymbol{D}^+)^\top$.

**Lemma 2.** *Define the loss of projecting a sample $\boldsymbol{x}$ with $\boldsymbol{D}$ or $\hat{\boldsymbol{H}}$ as in (16), (17). Then, we can write the loss difference as $\mathcal{L}(\hat{\boldsymbol{H}}) - \mathcal{L}(\boldsymbol{D}) = \mathbb{E}\left[\|\boldsymbol{z}\|_2^2 - \|\hat{\boldsymbol{z}}\|_2^2\right]$ and formulate it as*

$$\mathcal{L}(\hat{\boldsymbol{H}}) - \mathcal{L}(\boldsymbol{D}) = \sum_{i=1}^{\min(d,\hat{d})}\sum_{j=1}^m (\hat{\boldsymbol{h}}_i^\top \boldsymbol{h}_j)^2 (s_i - s_j) + \underbrace{\sum_{i=\hat{d}}^d s_i}_{=0 \text{ for } \hat{d} \geq d} + \underbrace{\sum_{i=d}^{\hat{d}}\sum_{j=1}^m (\hat{\boldsymbol{h}}_i^\top \boldsymbol{h}_j)^2 s_j}_{=0 \text{ for } \hat{d} \leq d}. \tag{18}$$

The result indicates that if we have perfect encoding ($\hat{d} = d$), then only the first term remains. If also all eigenvalues are equal, then there is no loss difference and the estimation of the direction of eigenvectors $\hat{\boldsymbol{H}}$ does not matter since we are dealing with the isotropic case. However, for more natural scenarios such as exponentially decaying eigenvalues, the eigenvalue difference is nonzero and correct estimation of the eigenvectors $\hat{\boldsymbol{H}}$ is crucial for a small loss difference. If we are dealing with imperfect encoding, there is either an additional term due to misalignment of the estimated eigenvalues ($\hat{d} < d$) or due to encoding of noise ($\hat{d} > d$). The proof is in Appendix F.1.

**Theorem 2.** *Define a real $t > 0$, using Corollary 4.1 from Loukas (2017), and with $k_j^2 = s_j(s_j + \text{Tr}(\boldsymbol{C}))$ from Corollary 4.3 in Loukas (2017), then we obtain the concentration inequality*

$$P\left(\mathcal{L}(\hat{\boldsymbol{H}}) - \mathcal{L}(\boldsymbol{D}) > t\right) \leq$$

$$\leq \frac{4}{t\, n_p}\left(\sum_{i=1}^{\min(d,\hat{d})}\sum_{j=i+1}^m \frac{k_j^2}{|s_i - s_j|} + \sum_{i=\hat{d}}^d\sum_{j=1}^m \frac{k_j^2 s_i}{(s_i - s_j)^2} + \sum_{i=d}^{\hat{d}}\sum_{j=1}^m \frac{k_j^2 s_j}{(s_i - s_j)^2}\right). \tag{19}$$

This theorem states, that in addition to the implications of Lemma 2, there are two main scenarios where we obtain a lower right hand side and therefore tighter bound. 1) When the feature covariance matrix has rapidly decaying eigenvalues, i.e. large $|s_i - s_j| \geq 0$, since $j > i$ or 2) when we have access to more pre-training samples $n_p$. The proof is in Appendix F.2.

**Connection to the risk**  We define the risk between features $x$ and outcomes $y$ in the same way as for the supervised case, see Lemma 1. The goal is to obtain asymptotic results for the risk for different eigenvalue decays including our latent variable data generator. Xu & Hsu (2019) presents asymptotic results for polynomial and more general eigenvalue decays in the PCA-regression model. However, their analysis relies on the assumption that the eigenvectors are fully known, i.e. $\hat{H}^{\top} = D^{+}$, which is an unrealistic scenario.

A solution to resolve this condition is to estimate the eigenvectors $\hat{H}$ from unlabeled data $\{x_i\}$. But it is unclear under what conditions the estimate is sufficiently good. The eigenvector estimation is precisely what is done during the pre-training step. Theorem 2 provides a sample complexity for the eigenvector estimation quality and therefore provides the missing condition when the results from Xu & Hsu (2019) hold in practice. Choosing $t$ sufficiently small, we can quantify how many samples are necessary for the estimated eigenvectors to be close to the true ones. Hence, we provide conditions when the asymptotic risk results from Xu & Hsu (2019) can be used in practice.

However, if we do not have access to sufficiently many pre-training data samples, then we know that our estimated eigenvectors $\hat{H}$ are misaligned. These eigenvectors will project the features into a misaligned latent space $\hat{z}$. Finally, we perform linear regression from this misaligned space. Quantifying the additional error on the overall risk for misaligned linear regression is an open problem.

### 5.3   Numerical results

We present numerical results when using pre-training. We denote the relation of pre-training samples to training samples as $\mu = \frac{n_p}{n}$ with $\mu \geq 1$ as we could use the training data set also for pre-training. We choose $d = 20$, $r_{\theta}^2 = 1$, $\sigma_y^2 = 0$, $\rho_x = 1$ and focus on $\hat{d} = d$ as the effects of misspecified models is equal as without pre-training and is elaborated in Section 4.2. Experiments to confirm this behavior for pre-training are in Appendix G. We generate $n = 200$ training samples and $n_p = n\mu$ pre-training samples by varying $\mu \in [1, 10]$ and average the computed risk over 100 realizations.

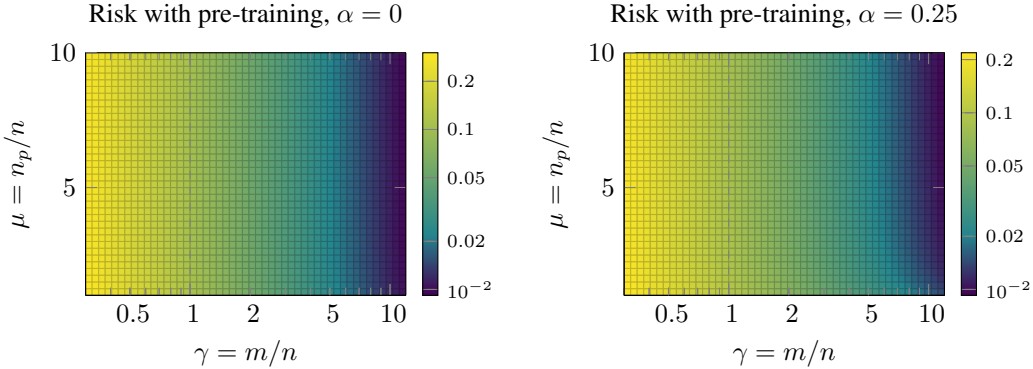

Figure 5: **Pre-training risk on latent variable data: simulation.** On the x-axis we increase the number of features $m$ and therefore the degree of overparameterization $\gamma$. On the y-axis we increase amount of pre-training $n_p$ compared to training data $n$. *Left:* Risk for latent variable data generated with feature covariance eigenvalue decay of $\alpha = 0$. *Right:* Same setup but for $\alpha = 0.25$.

In Figure 5 the risk for data with two different eigenvalue decay rates are depicted. We make three main observations: 1) Horizontally for $\mu = $ const., in both plots the risk decreases similar to the supervised case in Figure 3 and therefore follows Lemma 1. 2) Vertically for $\gamma = $ const., in the right plot ($\alpha = 0.25$) the risk decreases as expected from Theorem 2 when using more pre-training samples. The effect is most significant for large overparameterization $\gamma$. 3) Vertically for $\gamma = $ const., in the left plot ($\alpha = 0$), we notice that more pre-training data does *not* decrease the risk. Since we

have perfect encoding $\hat{d} = d$ and two blocks of constant eigenvalues, the eigenvector estimation is by Lemma 2 almost perfect and barely improves with more pre-training, see Theorem 2. Therefore, using more pre-training data does also *not* improve the overall risk. The observation supports our finding that more pre-training data decreases the risk only if it improves the eigenvector estimation. Hence, the eigenvalue distribution is crucial for the necessity of pre-training.

Figure 6 shows horizontal slices of Figure 5 (right) and compares it with fully supervised models. We notice that all pre-trained models outperform the fully supervised models for $\gamma > 1$. Interestingly, in the results for $\mu = 1$ (blue '×') we use the same amount of data to learn the PCA $n_p = n$ as in the fully supervised case (black triangles). While for the pre-trained model we use a different data set of the same size to learn the regression, we use use the exact same data in the supervised case.

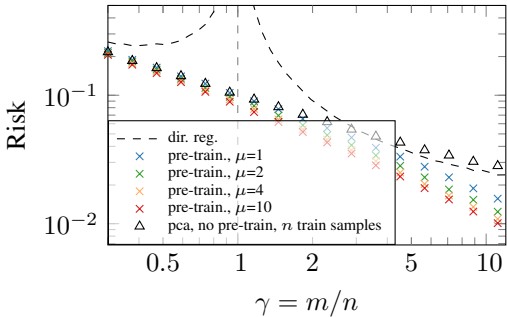

Figure 6: **Pre-training risk for different $\mu$: simulation.** Comparing horizontal slices of Figure 5 (right, $\alpha = 0.25$) for pre-trained models with different amount of pre-training data $\mu$ to 1) a fully supervised direct regression model and 2) a fully supervised PCA-regression model, comparable to Figure 3.

## 6 CONCLUSION

**Limitations**  Our proofs in the supervised case rely on random matrix theory for which we present asymptotic results for isotropic data. However, it is not trivial to find solutions for the general case, including our latent variable data generator, which requires more research. Similarly, it is an open question how to obtain a closed form solution for the complete risk in the scenario with pre-training based on eigenvector alignment which leads to our sample complexity bounds. Furthermore, while we observe key phenomena for our real-world example, the data here is not approximately on a low-dimensional manifold as our latent variable data generator and hence not fully comparable.

**Supervised case**  Our theoretical analysis generalizes the results for linear regression (Hastie et al., 2022) which is a special case of PCA-regression without dimensionality reduction ($\hat{d} = m$). In the non-asymptotic regime, Huang et al. (2022) describe similar results and hence they independently support our theory. Selecting the correct latent dimension $\hat{d}$ for data from a low dimensional manifold is crucial for the risk as Lemma 2 suggests. This is in line with the discovery of latent factors in variational autoencoders from the disentanglement literature (Higgins et al., 2017; Kumar et al., 2018). While our results that PCA mitigates the "interpolation peak" due to its regularizing behavior may not surprise, they provide formal guarantees for the performance of a commonly used model on real-world data structures. Practitioners can now rely on these fundamental guarantees for model development, but more research is needed for general data structures.

**Pre-training**  Our results from Figure 5 that a certain decay rate of the data covariance eigenvalues is necessary for pre-training to have its expected effect (more pre-training data is better) may be surprising at first. However, from Theorem 2 it becomes clear that more pre-training data only helps to improve the eigenvector estimation. If however, the eigenvectors are already estimated perfectly such as for two blocks of isotropic data (e.g. latent variable data with decay rate $\alpha = 0$), then using more pre-training data has no effect. Hence, we provide a fundamental insight into the mechanisms of pre-training which highlight that we have to be aware of the data structure instead of following the general philosophy of adding more pre-training data. Our results provide the missing link to Xu & Hsu (2019) when their asymptotic generalisation results can be used in practice. We believe that our simple PCA-regression model is suitable for extensive studies of pre-training phenomena. Therefore, this study lays the groundwork for future research and opens up many questions.

REPRODUCIBILITY

Code for reproducibility is attached as Jupyter notebook in the supplementary material and will be published online upon acceptance; all simulation parameters are explained in detail in the paper and copied in the code. All of our numerical simulations are run on Intel Core i7-6850K CPUs @ 3.60GHz in a matter of minutes. The computationally most heavy experiment is for pre-training with large $\mu$, see Figure 5 which takes for one run about 15 minutes for the fine-grained grid that we show in the paper. Averaging over multiple runs for more accurate results increase the computational cost linearly.

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

# Appendix

CONTENTS

## A    ADDITIONAL RELATED WORK

This section complements the related work in Section 2.

**Overparameterization—additional related work**    While the double descent has been observed in deep and state-of-the-art models (d'Ascoli et al., 2020; Nakkiran et al., 2021), most theoretical studies focus on simple models: Examples are found for linear regression (Bartlett et al., 2020; Muthukumar et al., 2020; Hastie et al., 2022), ensembles (LeJeune et al., 2020; Loureiro et al., 2021), classification (Gerace et al., 2020; Wang et al., 2021; Deng et al., 2022), random features (Belkin et al., 2019; Mei & Montanari, 2022) or small neural networks trained using gradient descent (Goldt et al., 2019; Advani et al., 2020).

**PCA-regression in applications**    The PCA-regression model is also known as principle component regression (PCR) (Xu & Hsu, 2019) or PCA-OLS (Huang et al., 2022). The number of chosen principal components or eigenvectors $\hat{d}$ is subject to model selection, see for example Xu & Hsu (2019) for an analysis if the true feature covariance matrix is fully known. Selecting $\hat{d}$ is a crucial step. While we do not specify how to select $\hat{d}$, we discuss the implication of model misspecification with $\hat{d} \neq d$. When using a *supervised* setup as in Section 4, there are plenty examples when it comes to use of PCA-regression models: Early work use the de-correlating property of PCA for their inputs in small scale examples (Massy, 1965). Tran et al. (2018) uses 10 years of data from Seoul to analyse the impact of air pollution on the health of the population using PCA-regression. Wang & Abbott (2008) makes use of PCA-regression for genetic association to determine genetic variants of human diseases with a large number of features and few samples. Metwally (2008) uses the model for spectrophotometry. When using *pre-training* as in Section 5, the PCA-regression model is a simplified, linear surrogate for large, nonlinear encoder-decoder models. Examples in this setting are is the transformer based BERT model (Devlin et al., 2019) or DALL-E Ramesh et al. (2021). In these models, parts of the model are pre-trained on a large corpus on unlabeled data. The pre-trained model can then be used by other developers to fine-tune or adapt the last layer, see Kumar et al. (2022).

## B EIGENVALUE DISTRIBUTION OF REAL-WORLD DATA SETS

In Figure 7 we plot the eigenvalue distribution of four real-world data sets. Each of them has a low number of significant eigenvalues with a sharp exponential decay. For some data sets such as e.g. Steel Plates Fault there is even a low-dimensional data embedding up to about eigenvalue 12 visible. All data sets except MNIST were downloaded from the UCI Machine Learning Repository (Dua & Graff, 2017) through the OpenML interface.

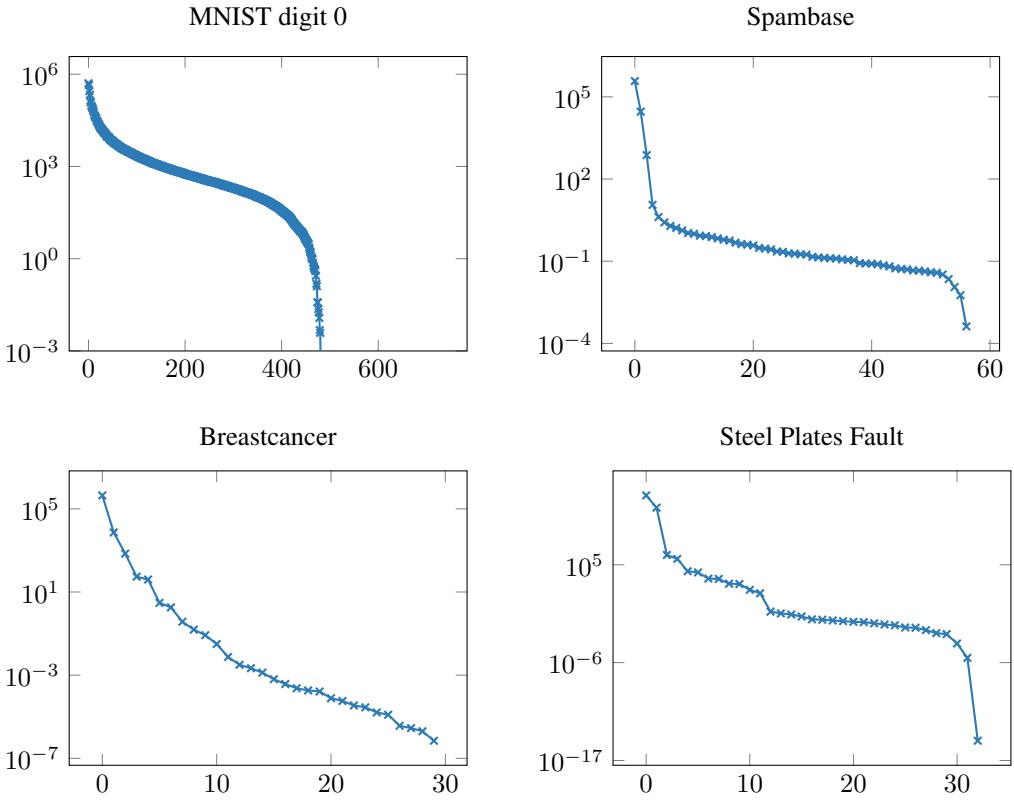

Figure 7: **Eigenvalue distribution of real-world data sets.** *Top left:* Distribution for MNIST digit 0 of the test data set (LeCun et al., 2010). Notice that many of the 784 eigenvalues are almost zero. *Top right:* Complete features of the spambase data set from UCI (Dua & Graff, 2017). *Bottom left:* Breastcancer data set from UCI (Dua & Graff, 2017). *Bottom right:* Steel-plates-fault data set provided by Semeion, Research of Sciences of Communication, Via Sersale 117, 00128, Rome, Italy.

## C  DETAILS ON THE DATA GENERATOR

In this appendix we concentrate without loss of generality on the latent variable data generator with orthogonal features introduced in (2) and (13). In matrix notation when collecting all features in rows we can write the data generator as

$$X = ZD^\top + ED_\perp^\top, \tag{20}$$
$$y = Z\theta + \varepsilon. \tag{21}$$

**Singular value and eigenvalue decomposition**  Approximating the data matrix with an estimated singular values decomposition and reducing the rank to $\hat{d}$ yields

$$X = \hat{U}\hat{\Sigma}\hat{V}^\top, \tag{22}$$

with estimated singular values $\hat{\Sigma} = \text{diag}(\hat{\sigma}_1, \ldots, \hat{\sigma}_d)$. Similarly we can define the eigenvalue decomposition of the sample covariance matrix as

$$\hat{C} = \frac{1}{n}X^\top X = \frac{1}{n}\hat{V}\hat{\Sigma}^\top\hat{\Sigma}\hat{V}^\top = \hat{V}\hat{S}\hat{V}^\top, \tag{23}$$

with estimated eigenvalue matrix $S = \text{diag}(\hat{s}_1, \ldots, \hat{s}_d)$.

**Covariance matrices**  The feature covariance matrix can be written as

$$C = \mathbb{E}\left[X^\top X\right] = [D \quad D_\perp]\mathbb{E}\left[\begin{bmatrix} Z^\top Z & Z^\top E \\ E^\top Z & E^\top E \end{bmatrix}\right]\begin{bmatrix} D^\top \\ D_\perp^\top \end{bmatrix} = VSV^\top, \tag{24}$$

where $V := [D \quad D_\perp]$ are the true eigenvectors—compare with the sample eigenvectors denoted by $\hat{V}$. The eigenvalue matrix $S$ can be written as

$$S = \text{diag}(s_1, \ldots, s_m) = \text{diag}(\lambda_1, \ldots, \lambda_d, 1, \ldots, 1) = \begin{bmatrix} \Lambda & 0 \\ 0 & I_{m-d} \end{bmatrix}. \tag{25}$$

**Signal-to-noise ratio control**  For the orthogonal latent variable data generator we can compute the SNR $\rho_x$ of the features as

$$\rho_x = \frac{\mathbb{E}\left[\|Dz\|_2^2\right]}{\mathbb{E}\left[\|ED_\perp\|_2^2\right]} = \frac{\text{Tr}(D\Lambda D^\top)}{\text{Tr}(D_\perp D_\perp^\top)} = \frac{\text{Tr}(c^2\Lambda)}{m-d}, \tag{26}$$

since $\text{Tr}(DD^\top) = \text{Tr}(I_{m-d})$ and $D^\top D = c^2 I_d$. Here $c$ is a correction factor which controls the SNR. We define is as

$$c = \sqrt{\frac{\rho_x(m-d)}{d}}\sqrt{\frac{d}{\text{Tr}(\Lambda)}}. \tag{27}$$

If non-orthogonal noise is used, then the first factor reduces to $\sqrt{\rho_x m/d}$.

In the same way, we can compute the SNR $\rho_y$ of the outputs as

$$\rho_y = \frac{\mathbb{E}\left[\|\theta^\top z\|_2^2\right]}{\mathbb{E}\left[\|\varepsilon\|_2^2\right]} = \frac{\text{Tr}(\theta^\top \Lambda\theta)}{\sigma_y^2} = \frac{r_\theta^2}{\sigma_y^2}, \tag{28}$$

with $r_\theta^2 = 1$ usually.

**Implementation details for data generation**  For our latent variable orthogonal feature generator, we generate the matrices $D$ and $D_\perp$ by first sampling an auxiliary random variable $A \in \mathbb{R}^{m \times d}$ and then orthogonalizing it with a QR-decomposition

$$A_{i,j} \sim \mathcal{N}(0,1), \tag{29}$$
$$QR = A, \tag{30}$$

where the columns of $\boldsymbol{Q}$ are orthogonal. Hence, we can define

$$\boldsymbol{D} = c\boldsymbol{Q}_{:d,:}, \tag{31}$$

$$\boldsymbol{D}_\perp = \boldsymbol{Q}_{d:,:}, \tag{32}$$

where the SNR-correction factor $c$ is defined in (27).

In order to hold that $\mathbb{E}\left[\|\boldsymbol{\theta}^\top \boldsymbol{z}\|_2^2\right] = r_{\boldsymbol{\theta}}^2$, we generate $\boldsymbol{\theta}$ as

$$\boldsymbol{\theta} = \frac{\sqrt{r_{\boldsymbol{\theta}}^2 d}}{\sqrt{d \operatorname{Tr}(\boldsymbol{\Lambda})}} \begin{bmatrix} 1 \\ \vdots \\ 1 \end{bmatrix}_d . \tag{33}$$

## D    PROOFS FOR THE SUPERVISED CASE

In this appendix we detail and proof the results shown in section 4.1. After stating some notation and definition in D.1, we first show the result for the linear regression solution in D.2. Subsequently, we will derive the result for the parameter norm in D.3 and its asymptotic in the isotropic case in D.4. Finally, the result for the risk is proven in D.5 and its asymptotic in the isotropic case is derived in D.6. For simplicity of notation, we will replace in the indices $\hat{d}$ by $d$–it is clear from context that we use $\hat{d}$ for the estimated latent space and $d$ for the true one.

### D.1    GENERAL

**Note on notation in main text**    While in the main paper for the derivation of the linear regression solution in (8) we denote for simplicity the estimated singular value matrix of the data $\boldsymbol{X}$ as $\hat{\boldsymbol{\Sigma}}$, here we are more precise. We distinguish between the case of $\gamma > 1$ with $m > n$ (left) and the case of $\gamma < 1$ with $m < n$ (right)

$$\hat{\boldsymbol{\Sigma}} = \left[ \begin{array}{ccc|c} \hat{\sigma}_1 & & \mathbf{0} & \\ & \ddots & & \mathbf{0} \\ \mathbf{0} & & \hat{\sigma}_n & \end{array} \right] \in \mathbb{R}^{n \times m}, \quad \hat{\boldsymbol{\Sigma}} = \left[ \begin{array}{ccc} \hat{\sigma}_1 & & \mathbf{0} \\ & \ddots & \\ \mathbf{0} & & \hat{\sigma}_m \\ \hline & \mathbf{0} & \end{array} \right] \in \mathbb{R}^{n \times m}. \tag{34}$$

When truncating with $d \ll \min(m, n)$ singular values, we obtain in both cases

$$\hat{\boldsymbol{\Sigma}}_d = \left[ \begin{array}{ccc} \hat{\sigma}_1 & & \mathbf{0} \\ & \ddots & \\ \mathbf{0} & & \hat{\sigma}_d \\ \hline & \mathbf{0}_{n-d \times d} & \end{array} \right] \in \mathbb{R}^{n \times \hat{d}}. \tag{35}$$

In (8) we write $\hat{\boldsymbol{\Sigma}}$ instead of $\hat{\boldsymbol{\Sigma}}_d$ and $\hat{\boldsymbol{\Sigma}}^{-1}$ instead of $(\hat{\boldsymbol{\Sigma}}_d^\top \hat{\boldsymbol{\Sigma}}_d)^{-1} \hat{\boldsymbol{\Sigma}}_d^\top$ in order to not overload notation and simplify reading without harming the results. In the following we will use the notation with subscript in order to highlight the zero rows or columns. Further, below we simplify the square matrix with only the first $\hat{d}$ singular values on the diagonal as $\hat{\boldsymbol{\Sigma}}_{dd}$ to indicate its dimensions.

**Sample covariance matrix**    We can define the feature sample covariance matrix $\hat{\boldsymbol{C}} \in \mathbb{R}^{m \times m}$ and its Moore-Penrose pseudoinverse as

$$\hat{\boldsymbol{C}} = \frac{1}{n} \boldsymbol{X}^\top \boldsymbol{X}, \qquad \hat{\boldsymbol{C}}^+ = n(\boldsymbol{X}^\top \boldsymbol{X})^+. \tag{36}$$

Using the definition of the truncated SVD, we can rewrite the sample covariance as

$$\hat{\boldsymbol{C}} = \frac{1}{n} \hat{\boldsymbol{V}} \hat{\boldsymbol{\Sigma}}_d^\top \hat{\boldsymbol{U}}^\top \hat{\boldsymbol{U}} \hat{\boldsymbol{\Sigma}}_d \hat{\boldsymbol{V}}^\top = \frac{1}{n} \hat{\boldsymbol{V}} \hat{\boldsymbol{\Sigma}}_d^\top \hat{\boldsymbol{\Sigma}}_d \hat{\boldsymbol{V}}^\top = \frac{1}{n} \hat{\boldsymbol{V}} \hat{\boldsymbol{\Sigma}}_{dd}^2 \hat{\boldsymbol{V}}^\top = \hat{\boldsymbol{V}} \hat{\boldsymbol{S}} \hat{\boldsymbol{V}}^\top, \tag{37}$$

$$\hat{\boldsymbol{C}}^+ = n(\hat{\boldsymbol{V}} \hat{\boldsymbol{\Sigma}}_d^\top \hat{\boldsymbol{\Sigma}}_d \hat{\boldsymbol{V}}^\top)^+ = n\hat{\boldsymbol{V}}(\hat{\boldsymbol{\Sigma}}_d^\top)^+ \hat{\boldsymbol{\Sigma}}_d^+ \hat{\boldsymbol{V}}^\top = n\hat{\boldsymbol{V}} \hat{\boldsymbol{\Sigma}}_{dd}^{-2} \hat{\boldsymbol{V}}^\top = \hat{\boldsymbol{V}} \hat{\boldsymbol{S}}^{-1} \hat{\boldsymbol{V}}^\top. \tag{38}$$

The above formulation also implies the following which will be useful

$$\hat{\boldsymbol{\Sigma}}_{dd}^{-2} = \frac{1}{n} \hat{\boldsymbol{V}}^\top \hat{\boldsymbol{C}}^+ \hat{\boldsymbol{V}}. \tag{39}$$

### D.2    LINEAR REGRESSION

We consider the unregularized linear regression solution between the latent variables $\hat{\boldsymbol{Z}}$ and the outcome $\boldsymbol{y}$:

$$\hat{\boldsymbol{\theta}} = (\hat{\boldsymbol{Z}}^\top \hat{\boldsymbol{Z}})^+ \hat{\boldsymbol{Z}}^\top \boldsymbol{y} \tag{40}$$

$$= (\hat{\boldsymbol{\Sigma}}_d^\top \hat{\boldsymbol{U}}^\top \hat{\boldsymbol{U}} \hat{\boldsymbol{\Sigma}}_d)^+ \hat{\boldsymbol{\Sigma}}_d^\top \hat{\boldsymbol{U}}^\top \boldsymbol{y} \tag{41}$$

with $\hat{\boldsymbol{U}}^\top \hat{\boldsymbol{U}} = \boldsymbol{I}$ and $\boldsymbol{y} = \boldsymbol{X} \boldsymbol{\beta} + \boldsymbol{\epsilon}$

$$\hat{\boldsymbol{\theta}} = (\hat{\boldsymbol{\Sigma}}_d^\top \hat{\boldsymbol{\Sigma}}_d)^+ \hat{\boldsymbol{\Sigma}}_d^\top \hat{\boldsymbol{U}}^\top (\boldsymbol{X} \boldsymbol{\beta} + \boldsymbol{\epsilon}) \tag{42}$$

$$= (\hat{\boldsymbol{\Sigma}}_d^\top \hat{\boldsymbol{\Sigma}}_d)^+ \hat{\boldsymbol{\Sigma}}_d^\top \hat{\boldsymbol{\Sigma}} \hat{\boldsymbol{V}}^\top \boldsymbol{\beta} + (\hat{\boldsymbol{\Sigma}}_d^\top \hat{\boldsymbol{\Sigma}}_d)^+ \hat{\boldsymbol{\Sigma}}_d^\top \hat{\boldsymbol{U}}^\top \boldsymbol{\epsilon} \tag{43}$$

Where we used $\boldsymbol{X} = \hat{\boldsymbol{U}}\hat{\boldsymbol{\Sigma}}\hat{\boldsymbol{V}}^\top$. Now we combine the singular value matrices. We indicate dimensions of combined matrices. Note that $\hat{\boldsymbol{\Sigma}}_d$ and $\hat{\boldsymbol{\Sigma}}$ are of different sizes.

$$
\hat{\boldsymbol{\theta}} = \begin{bmatrix} \frac{1}{\hat{\sigma}_1^2} & & \mathbf{0} \\ & \ddots & \\ \mathbf{0} & & \frac{1}{\hat{\sigma}_d^2} \end{bmatrix}_{d\times d} \begin{bmatrix} \hat{\sigma}_1^2 & & \mathbf{0} & \\ & \ddots & & \mathbf{0} \\ \mathbf{0} & & \hat{\sigma}_d^2 & \end{bmatrix}_{d\times m} \hat{\boldsymbol{V}}^\top \boldsymbol{\beta} \;+\;
$$
$$
+ \begin{bmatrix} \frac{1}{\hat{\sigma}_1} & & \mathbf{0} & \\ & \ddots & & \mathbf{0} \\ \mathbf{0} & & \frac{1}{\hat{\sigma}_d} & \end{bmatrix}_{d\times n} \hat{\boldsymbol{U}}^\top \boldsymbol{\epsilon}
\tag{44}
$$

$$
= [\boldsymbol{I}_d \quad \mathbf{0}]\,\hat{\boldsymbol{V}}^\top \boldsymbol{\beta} + [\hat{\boldsymbol{\Sigma}}_{dd}^{-1} \quad \mathbf{0}]\,\hat{\boldsymbol{U}}^\top \boldsymbol{\epsilon}
\tag{45}
$$

Summarizing the matrices by truncating $\hat{\boldsymbol{V}}^\top$ and $\hat{\boldsymbol{U}}^\top$ yields the following solution for the regression parameter estimation

$$
\boxed{\hat{\boldsymbol{\theta}} = \hat{\boldsymbol{V}}_d^\top \boldsymbol{\beta} + \hat{\boldsymbol{\Sigma}}_{dd}^{-1}\hat{\boldsymbol{U}}_d^\top \boldsymbol{\epsilon}}
\tag{46}
$$

### D.3 PARAMETER NORM

Here, we prove the parameter norm part of Lemma 1.

*Proof.* In order to evaluate the parameter norm $\|\hat{\boldsymbol{\theta}}\|_2^2 = \hat{\boldsymbol{\theta}}^\top\hat{\boldsymbol{\theta}}$, we consider

$$
\hat{\boldsymbol{\theta}}^\top\hat{\boldsymbol{\theta}} = \boldsymbol{\beta}^\top \hat{\boldsymbol{V}}\hat{\boldsymbol{V}}^\top \boldsymbol{\beta} + \mathrm{Tr}(\boldsymbol{\epsilon}^\top \hat{\boldsymbol{U}}\hat{\boldsymbol{\Sigma}}_{dd}^{-1}\hat{\boldsymbol{\Sigma}}_{dd}^{-1}\hat{\boldsymbol{U}}^\top \boldsymbol{\epsilon}) + 2\boldsymbol{\beta}^\top \hat{\boldsymbol{V}}\hat{\boldsymbol{\Sigma}}_{dd}^{-1}\hat{\boldsymbol{U}}^\top \boldsymbol{\epsilon}
\tag{47}
$$

where the second term is scalar and hence equal to its trace. Now, we can make use of the cyclic property of the trace. Furthermore, define $\boldsymbol{\Phi} := \hat{\boldsymbol{V}}\hat{\boldsymbol{V}}^\top$ as an orthogonal projector.

$$
\hat{\boldsymbol{\theta}}^\top\hat{\boldsymbol{\theta}} = \boldsymbol{\beta}^\top \boldsymbol{\Phi}\boldsymbol{\beta} + \mathrm{Tr}(\hat{\boldsymbol{U}}^\top \boldsymbol{\epsilon}\boldsymbol{\epsilon}^\top \hat{\boldsymbol{U}}\hat{\boldsymbol{\Sigma}}_{dd}^{-2}) + 2\boldsymbol{\beta}^\top \hat{\boldsymbol{V}}\hat{\boldsymbol{\Sigma}}_{dd}^{-1}\hat{\boldsymbol{U}}^\top \boldsymbol{\epsilon}
\tag{48}
$$

Note that the properties of the orthogonal projector with $\boldsymbol{\Phi}^\top = \boldsymbol{\Phi}$ and $\boldsymbol{\Phi}\boldsymbol{\Phi} = \boldsymbol{\Phi}$ hold for our definition.

We take the expectation with respect to the noise

$$
\mathbb{E}_{\boldsymbol{\epsilon}}\left[\hat{\boldsymbol{\theta}}^\top\hat{\boldsymbol{\theta}}\right] = \boldsymbol{\beta}^\top \boldsymbol{\Phi}\boldsymbol{\beta} + \mathrm{Tr}(\hat{\boldsymbol{U}}^\top \mathbb{E}_{\boldsymbol{\epsilon}}\left[\boldsymbol{\epsilon}\boldsymbol{\epsilon}^\top\right]\hat{\boldsymbol{U}}\hat{\boldsymbol{\Sigma}}_{dd}^{-2})
\tag{49}
$$

$$
= \boldsymbol{\beta}^\top \boldsymbol{\Phi}\boldsymbol{\beta} + \sigma_\varepsilon^2 \,\mathrm{Tr}(\hat{\boldsymbol{U}}^\top \hat{\boldsymbol{U}}\hat{\boldsymbol{\Sigma}}_{dd}^{-2})
\tag{50}
$$

$$
= \boldsymbol{\beta}^\top \boldsymbol{\Phi}\boldsymbol{\beta} + \sigma_\varepsilon^2 \,\mathrm{Tr}(\hat{\boldsymbol{\Sigma}}_{dd}^{-2})
\tag{51}
$$

using (39) we can write

$$
\boxed{\mathbb{E}_{\boldsymbol{\epsilon}}\left[\hat{\boldsymbol{\theta}}^\top\hat{\boldsymbol{\theta}}\right] = \boldsymbol{\beta}^\top \boldsymbol{\Phi}\boldsymbol{\beta} + \frac{\sigma_\epsilon^2}{n} \,\mathrm{Tr}(\hat{\boldsymbol{V}}^\top \hat{\boldsymbol{C}}^+\hat{\boldsymbol{V}})}
\tag{52}
$$

The second term uses the sample covariance matrix $\hat{\boldsymbol{C}}$ projected down on the $\hat{d}$ dimensional eigenvector space using $\hat{\boldsymbol{V}}$. $\qquad\square$

### D.4 LIMITING PARAMETER NORM FOR ISOTROPIC FEATURES

Here, we prove the parameter norm part of Theorem 1.

*Proof.* We can analyze the two terms in (52) independently in the limit of $m, n \to \infty$ such that $\frac{m}{n} \to \gamma \in (0, \infty)$ almost surely. Furthermore we assume isotropic features $\mathrm{Cov}(\boldsymbol{x}_i) = \boldsymbol{C} = \boldsymbol{I}_m$.

**First term** We can write with the definition of our orthogonal projector.

$$\boldsymbol{\beta}^\top \boldsymbol{\Phi} \boldsymbol{\beta} = \boldsymbol{\beta}^\top \hat{\boldsymbol{V}} \hat{\boldsymbol{V}}^\top \boldsymbol{\beta} \tag{53}$$

We can write $\hat{\boldsymbol{V}}^\top$ with the SVD definition as $\hat{\boldsymbol{V}}^\top = \hat{\boldsymbol{\Sigma}}_d^+ \hat{\boldsymbol{U}}^\top \boldsymbol{X}$ which yields

$$\boldsymbol{\beta}^\top \boldsymbol{\Phi} \boldsymbol{\beta} = \boldsymbol{\beta}^\top \boldsymbol{X}^\top \hat{\boldsymbol{U}} \hat{\boldsymbol{\Sigma}}_d^{+\top} \hat{\boldsymbol{\Sigma}}_d^+ \hat{\boldsymbol{U}}^\top \boldsymbol{X} \boldsymbol{\beta} \tag{54}$$

For the special case of i.i.d. matrix entries $\boldsymbol{x}_i \sim \mathcal{N}(0,1)$ we have by rotational invariance that the distribution of $\boldsymbol{X}$ and $\boldsymbol{X}\boldsymbol{P}$ are equal for any orthogonal $\boldsymbol{P} \in \mathbb{R}^{m \times m}$

$$\boldsymbol{\beta}^\top \boldsymbol{\Phi} \boldsymbol{\beta} = \boldsymbol{\beta}^\top \boldsymbol{P}^\top \boldsymbol{X}^\top \hat{\boldsymbol{U}} \hat{\boldsymbol{\Sigma}}_d^{+\top} \hat{\boldsymbol{\Sigma}}_d^+ \hat{\boldsymbol{U}}^\top \boldsymbol{X} \boldsymbol{P} \boldsymbol{\beta} \tag{55}$$

Choose $\boldsymbol{P}$ such that $\boldsymbol{P}\boldsymbol{\beta} = \boldsymbol{\beta}\boldsymbol{e}_i$ with $\boldsymbol{e}_i$ as the $i$th normal vector and then average over $i = 1, \ldots, m$

$$\boldsymbol{\beta}^\top \boldsymbol{\Phi} \boldsymbol{\beta} = \boldsymbol{\beta}^\top \boldsymbol{\beta} \operatorname{Tr}(\boldsymbol{X}^\top \hat{\boldsymbol{U}} \hat{\boldsymbol{\Sigma}}_d^{+\top} \hat{\boldsymbol{\Sigma}}_d^+ \hat{\boldsymbol{U}}^\top \boldsymbol{X})/m \tag{56}$$

Now use again the definition of $\boldsymbol{X} = \hat{\boldsymbol{U}} \hat{\boldsymbol{\Sigma}} \hat{\boldsymbol{V}}^\top$ yields

$$\boldsymbol{\beta}^\top \boldsymbol{\Phi} \boldsymbol{\beta} = \boldsymbol{\beta}^\top \boldsymbol{\beta} \operatorname{Tr}(\hat{\boldsymbol{V}} \hat{\boldsymbol{\Sigma}}^\top \hat{\boldsymbol{U}}^\top \hat{\boldsymbol{U}} \hat{\boldsymbol{\Sigma}}_d^{+\top} \hat{\boldsymbol{\Sigma}}_d^+ \hat{\boldsymbol{U}}^\top \hat{\boldsymbol{U}} \hat{\boldsymbol{\Sigma}} \hat{\boldsymbol{V}}^\top)/m \tag{57}$$

$$= \boldsymbol{\beta}^\top \boldsymbol{\beta} \operatorname{Tr}(\hat{\boldsymbol{V}} \hat{\boldsymbol{\Sigma}}^\top \hat{\boldsymbol{\Sigma}}_d^{+\top} \hat{\boldsymbol{\Sigma}}_d^+ \hat{\boldsymbol{\Sigma}} \hat{\boldsymbol{V}}^\top)/m \tag{58}$$

Using the same arguments as for the linear regression parameter solution by combining the singular value matrices, we obtain

$$\boldsymbol{\beta}^\top \boldsymbol{\Phi} \boldsymbol{\beta} = \boldsymbol{\beta}^\top \boldsymbol{\beta} \operatorname{Tr}(\hat{\boldsymbol{V}} \hat{\boldsymbol{V}}^\top)/m \tag{59}$$

Here we again identify our orthogonal projector $\boldsymbol{\Phi}$

$$= \boldsymbol{\beta}^\top \boldsymbol{\beta} \operatorname{Tr}(\boldsymbol{\Phi})/m \tag{60}$$

Since $\boldsymbol{\Phi}$ is symmetric positive definite and since its components $\hat{\boldsymbol{V}}$ are orthogonal, all eigenvalues of $\boldsymbol{\Phi}$ are equal to one, yielding

$$\boldsymbol{\beta}^\top \boldsymbol{\Phi} \boldsymbol{\beta} = \boldsymbol{\beta}^\top \boldsymbol{\beta} \operatorname{rank}(\boldsymbol{\Phi})/m \tag{61}$$

For $m/n \to \gamma$ we have to distinguish between $\gamma < 1$ and $\gamma > 1$. Therefore, we obtain the final version for the first term of the limiting parameter norm:

$$\boxed{\boldsymbol{\beta}^\top \boldsymbol{\Phi} \boldsymbol{\beta} = \begin{cases} \boldsymbol{\beta}^\top \boldsymbol{\beta} \min(\hat{d}, m)/m & \text{for} \quad \gamma < 1 \\ \boldsymbol{\beta}^\top \boldsymbol{\beta} \min(\hat{d}, n)/m & \text{for} \quad \gamma > 1 \end{cases}} \tag{62}$$

Checking the results with considering all principal components, i.e. choosing $\hat{d} = m$ (with $m > n$ for $\gamma > 1$), we obtain

$$\boldsymbol{\beta}^\top \boldsymbol{\Phi} \boldsymbol{\beta} = \begin{cases} \boldsymbol{\beta}^\top \boldsymbol{\beta} & \text{for} \quad \gamma < 1 \\ \boldsymbol{\beta}^\top \boldsymbol{\beta} \frac{1}{\gamma} & \text{for} \quad \gamma > 1 \end{cases}$$

which is the same results as for the case of direct regression between $\boldsymbol{X}$ and $\boldsymbol{y}$.

**Second term** For the second term of the parameter norm we can write the trace as the sum over the eigenvalues $s_i$ of $\hat{\boldsymbol{C}}$ but limited to the first $\hat{d}$ eigenvalues due to the projection using $\hat{\boldsymbol{V}}$

$$\frac{\sigma_\epsilon^2}{n} \operatorname{Tr}(\hat{\boldsymbol{V}}^\top \hat{\boldsymbol{C}}^+ \hat{\boldsymbol{V}}) = \sigma_\epsilon^2 \frac{1}{n} \sum_{i=1}^{\hat{d}} \frac{1}{s_i} \tag{63}$$

$$= \sigma_\epsilon^2 \frac{m}{n} \int_{s_f}^\infty \frac{1}{s} dF_{\hat{C}}(s) \tag{64}$$

where the summation is rewritten as integral over the spectral measure $F_{\hat{C}}$ of $\hat{\boldsymbol{C}}$ as and $s_f$ is the $\hat{d}$ largest eigenvalue of $\hat{\boldsymbol{C}}$. We know that in the limit $m, n \to \infty$ the spectral measure will almost surely converge to the Marčenko-Pastur distribution $F_\gamma$ which describes the distribution of the eigenvalues of $\hat{\boldsymbol{C}}$

$$\boxed{\frac{\sigma_\epsilon^2}{n} \operatorname{Tr}(\hat{\boldsymbol{V}}^\top \hat{\boldsymbol{C}}^+ \hat{\boldsymbol{V}}) \to \sigma_\epsilon^2 \frac{m}{n} \int_{s_f}^\infty \frac{1}{s} dF_\gamma(s)} \tag{65}$$

There are now two steps to solve this integral. First, we need to find out the lower integral bound $s_f$ and second, solve the integral itself. For $s_f = -\infty$, one can use the closed form solution of the Stieltjes transformation $f(z)$ of the Marčenko-Pastur distribution and evaluate it at $z = 0$. However, there is no known closed form solution for general $s_f$. We therefore solve this part numerically.

*Step 1 obtain the lower bound $s_f$:* We can view the spectral measure as $F_{\hat{\Sigma}}$ as a series of $m$ impulses at $s_i$ with magnitude $1/m$ because the sum is normalized to 1. Since we only consider the $\hat{d}$ largest eigenvalues, we know that their sum is $\hat{d}/m$, see Figure 8a. This sum is the same as the integral from $s_f$ over the Marčenko-Pastur distribution, see Figure 8b. Therefore we can find the lower integral bound $s_f$ by solving

$$\frac{\hat{d}}{m} = \int_{s_f}^{\infty} dF_\gamma(s) \tag{66}$$

$$= \int_{s_f}^{s_+} \frac{1}{2\pi} \frac{\sqrt{(s_+ - s)(s - s_-)}}{\gamma s} ds \tag{67}$$

for $s_f$ numerically with $s_\pm = (1 \pm \sqrt{\gamma})^2$ where $s_\pm$ is the lowest/highest eigenvalue. Note that $s \in [s_-, s_+]$.

*Step 2 solve integral of interest:* Now we can solve the integral in (65) numerically from $s_f$ to the upper bound $s_+$. Therefore, we obtain a solution for the second term, which is not based on data but the properties of our data matrix, especially $\gamma$ and $\hat{d}$. This concludes the full proof for the asymptotics of the parameter norm. $\square$

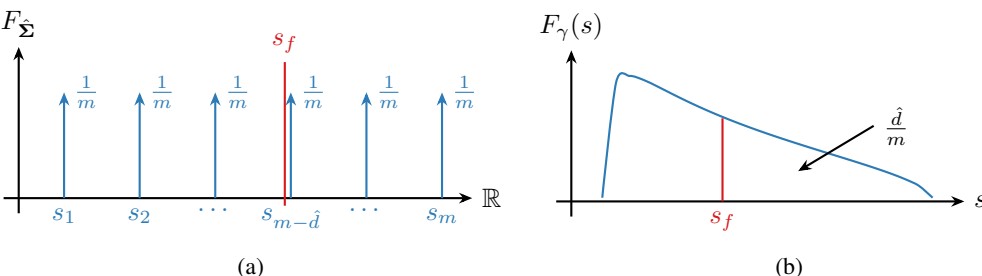

(a)                                            (b)

Figure 8: **Visualization of steps for variance term derivation**. *(a)* spectral measure impulses and and lower integral bound of integral $s_f$. *(b)* Marčenko-Pastur distribution (Marčenko & Pastur, 1967) for $\gamma = 0.3$ with specific lower integration bound. The area under the distribution from that threshold is equal to $\hat{d}/m$.

### D.5 RISK

Here, we prove the risk part of Lemma 1.

*Proof.* We define the risk as the expectation over the mean squared error, and then use $y_0 = \boldsymbol{\beta}^\top \boldsymbol{x}_0 + \epsilon$, $\hat{y}(\boldsymbol{x}_0) = \hat{\boldsymbol{\theta}}^\top \hat{\boldsymbol{z}}$ and $\hat{\boldsymbol{z}} = \hat{\boldsymbol{V}}^\top \boldsymbol{x}_0$ to obtain

$$R(\hat{\boldsymbol{\theta}}) = \mathbb{E}_{(\boldsymbol{x}_0, y_0)} \left[ (y_0 - \hat{y}(\boldsymbol{x}_0))^2 \right] \tag{68}$$

$$= \mathbb{E}_{\boldsymbol{x}_0} \left[ (\boldsymbol{\beta}^\top \boldsymbol{x}_0 + \epsilon - \hat{y}(\boldsymbol{x}_0))^2 \right] \tag{69}$$

$$= \mathbb{E}_{\boldsymbol{x}_0} \left[ (\boldsymbol{\beta}^\top \boldsymbol{x}_0 + \epsilon - \hat{\boldsymbol{\theta}}^\top \hat{\boldsymbol{z}})^2 \right] \tag{70}$$

$$= \mathbb{E}_{\boldsymbol{x}_0} \left[ (\boldsymbol{\beta}^\top \boldsymbol{x}_0 + \epsilon - \hat{\boldsymbol{\theta}}^\top \hat{\boldsymbol{V}}^\top \boldsymbol{x}_0)^2 \right] \tag{71}$$

$$= \mathbb{E}_{\boldsymbol{x}_0} \left[ ((\boldsymbol{\beta} - \hat{\boldsymbol{V}}\hat{\boldsymbol{\theta}})^\top \boldsymbol{x}_0 + \epsilon)^2 \right] \tag{72}$$

$$= (\boldsymbol{\beta} - \hat{\boldsymbol{V}}\hat{\boldsymbol{\theta}})^\top \boldsymbol{C} (\boldsymbol{\beta} - \hat{\boldsymbol{V}}\hat{\boldsymbol{\theta}}) + \epsilon \epsilon^\top \tag{73}$$

For simplicity we first rephrase the term in the bracket using the solution of the regression parameter estimation in (46). We re-use our orthogonal projector $\boldsymbol{\Phi} = \hat{\boldsymbol{V}}\hat{\boldsymbol{V}}^\top$ and define another orthogonal projector with $\boldsymbol{\Pi} = \boldsymbol{I}_m - \boldsymbol{\Phi}$ to obtain

$$\boldsymbol{\beta} - \hat{\boldsymbol{V}}\hat{\boldsymbol{\theta}} = \boldsymbol{\beta} - \hat{\boldsymbol{V}}(\hat{\boldsymbol{V}}^\top\boldsymbol{\beta} + \hat{\boldsymbol{\Sigma}}_{dd}^{-1}\hat{\boldsymbol{U}}^\top\epsilon) \tag{74}$$

$$= \boldsymbol{\beta} - \hat{\boldsymbol{V}}\hat{\boldsymbol{V}}^\top\boldsymbol{\beta} - \hat{\boldsymbol{V}}\hat{\boldsymbol{\Sigma}}_{dd}^{-1}\hat{\boldsymbol{U}}^\top\epsilon \tag{75}$$

$$= (\boldsymbol{I}_m - \boldsymbol{\Phi})\boldsymbol{\beta} - \hat{\boldsymbol{V}}\hat{\boldsymbol{\Sigma}}_{dd}^{-1}\hat{\boldsymbol{U}}^\top\epsilon \tag{76}$$

$$= \boldsymbol{\Pi}\boldsymbol{\beta} - \hat{\boldsymbol{V}}\hat{\boldsymbol{\Sigma}}_{dd}^{-1}\hat{\boldsymbol{U}}^\top\epsilon \tag{77}$$

Now we use this expression to take the expectation of the risk with respect to the noise. This yields

$$\mathbb{E}_\epsilon\left[R(\hat{\boldsymbol{\theta}})\right] = \boldsymbol{\beta}^\top\boldsymbol{\Pi}\boldsymbol{C}\boldsymbol{\Pi}\boldsymbol{\beta} + \mathbb{E}_\epsilon\left[\text{Tr}(\epsilon^\top\hat{\boldsymbol{U}}\hat{\boldsymbol{\Sigma}}_{dd}^{-1}\hat{\boldsymbol{V}}^\top\boldsymbol{C}\hat{\boldsymbol{V}}\hat{\boldsymbol{\Sigma}}_{dd}^{-1}\hat{\boldsymbol{U}}^\top\epsilon)\right] + \mathbb{E}_\epsilon\left[\epsilon\epsilon^\top\right] \tag{78}$$

Here we made use of the Trace since the expression is scalar. Hence, we can use the cyclic property of the trace and pull the expectation inside

$$= \boldsymbol{\beta}^\top\boldsymbol{\Pi}\boldsymbol{C}\boldsymbol{\Pi}\boldsymbol{\beta} + \text{Tr}(\hat{\boldsymbol{V}}^\top\boldsymbol{C}\hat{\boldsymbol{V}}\hat{\boldsymbol{\Sigma}}_{dd}^{-1}\hat{\boldsymbol{U}}^\top\mathbb{E}_\epsilon\left[\epsilon\epsilon^\top\right]\hat{\boldsymbol{U}}\hat{\boldsymbol{\Sigma}}_{dd}^{-1}) + \mathbb{E}_\epsilon\left[\epsilon\epsilon^\top\right] \tag{79}$$

with $\mathbb{E}_\epsilon\left[\epsilon\epsilon^\top\right] = \sigma_\epsilon^2$ and $\hat{\boldsymbol{U}}^\top\hat{\boldsymbol{U}} = \boldsymbol{I}$

$$= \boldsymbol{\beta}^\top\boldsymbol{\Pi}\boldsymbol{C}\boldsymbol{\Pi}\boldsymbol{\beta} + \sigma_\epsilon^2\text{Tr}(\hat{\boldsymbol{V}}^\top\boldsymbol{C}\hat{\boldsymbol{V}}\hat{\boldsymbol{\Sigma}}_{dd}^{-2}) + \sigma_\epsilon^2 \tag{80}$$

using (39) for $\hat{\boldsymbol{\Sigma}}_{dd}^{-2}$

$$\boxed{= \boldsymbol{\beta}^\top\boldsymbol{\Pi}\boldsymbol{C}\boldsymbol{\Pi}\boldsymbol{\beta} + \frac{\sigma_\epsilon^2}{n}\text{Tr}(\hat{\boldsymbol{V}}^\top\boldsymbol{C}\hat{\boldsymbol{V}}\hat{\boldsymbol{V}}^\top\hat{\boldsymbol{C}}^+\hat{\boldsymbol{V}}) + \sigma_\epsilon^2} \tag{81}$$

Again, similarly to the parameter norm, the second term here uses the covariance matrices projected onto the $\hat{d}$ dimensional eigenvector space. $\qquad\square$

### D.6 Limiting risk for isotropic features

Here, we prove the risk part of Theorem 1.

*Proof.* Since we use isotropic features, we have $\boldsymbol{C} = \boldsymbol{I}_m$. Similar to the limiting parameter norm we split the analysis for the two first parts of (81).

**First term: limiting bias** Using isotopic features and the definition of the orthogonal projector, we have

$$\boldsymbol{\beta}^\top\boldsymbol{\Pi}\boldsymbol{C}\boldsymbol{\Pi}\boldsymbol{\beta} = \boldsymbol{\beta}^\top\boldsymbol{\Pi}\boldsymbol{\beta} \tag{82}$$

$$= \boldsymbol{\beta}^\top(\boldsymbol{I}_m - \hat{\boldsymbol{V}}\hat{\boldsymbol{V}}^\top)\boldsymbol{\beta} \tag{83}$$

now we can use the same arguments as for the first term in the parameter norm. Namely, rewrite $\hat{\boldsymbol{V}}^\top = \hat{\boldsymbol{\Sigma}}_d^+\hat{\boldsymbol{U}}^\top\boldsymbol{X}$ in terms of $\boldsymbol{X}$, assume $\boldsymbol{x}_i \sim \mathcal{N}(0, 1)$ and by rotation invariance the distribution of $\boldsymbol{X}$ and $\boldsymbol{X}\boldsymbol{P}$ are equal, where $\boldsymbol{P}$ is any orthogonal matrix. Then we choose $\boldsymbol{P}\boldsymbol{\beta} = \beta\boldsymbol{e}_i$ and average over all $i = 1, \ldots, m$.

$$= \boldsymbol{\beta}^\top\left(\boldsymbol{I}_m - \boldsymbol{P}^\top\boldsymbol{X}^\top\hat{\boldsymbol{U}}\hat{\boldsymbol{\Sigma}}_d^{+\top}\hat{\boldsymbol{\Sigma}}_d^+\hat{\boldsymbol{U}}^\top\boldsymbol{X}\boldsymbol{P}\right)\boldsymbol{\beta} \tag{84}$$

$$= \boldsymbol{\beta}^\top\boldsymbol{\beta}\left(1 - \text{Tr}(\boldsymbol{X}^\top\hat{\boldsymbol{U}}\hat{\boldsymbol{\Sigma}}_d^{+\top}\hat{\boldsymbol{\Sigma}}_d^+\hat{\boldsymbol{U}}^\top\boldsymbol{X})/m\right) \tag{85}$$

$$= \boldsymbol{\beta}^\top\boldsymbol{\beta}\left(1 - \text{Tr}(\hat{\boldsymbol{V}}\hat{\boldsymbol{V}}^\top)/m\right) \tag{86}$$

$$= \boldsymbol{\beta}^\top\boldsymbol{\beta}\left(1 - \text{Tr}(\boldsymbol{\Phi})/m\right) \tag{87}$$

$$= \boldsymbol{\beta}^\top\boldsymbol{\beta}\left(1 - \text{rank}(\boldsymbol{\Phi})/m\right) \tag{88}$$

in the limit of $m, n \to \infty$ we have $\frac{m}{n} \to \gamma$ almost surely. We therefore obtain the final solution for the limiting bias as

$$\boldsymbol{\beta}^\top \boldsymbol{\Pi C \Pi \beta} = \begin{cases} \boldsymbol{\beta}^\top \boldsymbol{\beta} \left(1 - \min(\hat{d}, m)/m\right) & \text{for} \quad \gamma < 1 \\ \boldsymbol{\beta}^\top \boldsymbol{\beta} \left(1 - \min(\hat{d}, n)/m\right) & \text{for} \quad \gamma > 1 \end{cases} \tag{89}$$

Again, checking the results with considering all principal components, i.e. choosing $\hat{d} = m$ (with $m > n$ for $\gamma > 1$), we obtain

$$\boldsymbol{\beta}^\top \boldsymbol{\Phi \beta} = \begin{cases} 0 & \text{for} \quad \gamma < 1 \\ \boldsymbol{\beta}^\top \boldsymbol{\beta} \left(1 - \frac{1}{\gamma}\right) & \text{for} \quad \gamma > 1 \end{cases}$$

which is the same results as for the case of direct regression between $\boldsymbol{X}$ and $\boldsymbol{y}$.

**Second term: limiting variance** Using isotropic features we have

$$\frac{\sigma_\epsilon^2}{n} \operatorname{Tr}(\hat{\boldsymbol{V}}^\top \boldsymbol{C} \hat{\boldsymbol{V}} \hat{\boldsymbol{V}}^\top \hat{\boldsymbol{C}}^+ \hat{\boldsymbol{V}}) = \frac{\sigma_\epsilon^2}{n} \operatorname{Tr}(\hat{\boldsymbol{V}}^\top \hat{\boldsymbol{C}}^+ \hat{\boldsymbol{V}}) \tag{90}$$

this is the same form as the second term for the parameter norm and therefore yields the same numeric solution by solving

$$\frac{\sigma_\epsilon^2}{n} \operatorname{Tr}(\hat{\boldsymbol{V}}^\top \boldsymbol{C} \hat{\boldsymbol{V}} \hat{\boldsymbol{V}}^\top \hat{\boldsymbol{C}}^+ \hat{\boldsymbol{V}}) = \sigma_\epsilon^2 \frac{m}{n} \int_{s_f}^\infty \frac{1}{s} dF_\gamma(s) \tag{91}$$

$\square$

# E    ADDITIONAL NUMERICAL RESULTS FOR SUPERVISED CASE

## E.1    ISOTROPIC DATA: BIAS-VARIANCE DECOMPOSITION

In Figure 9 we extend Figure 2. We additionally show the results for the PCA-regression model with
$\hat{d} = m$, which corresponds to a PCA without compression and therefore a direct regression between
input $x$ and output $y$. We compare the analytical results from Theorem 1 of the PCA-model to 1) the
direction regression model from simulations (solid line) and 2) the analytical solution for direct
regression of isotropic data from Hastie et al. (2022). The authors give the solution for the risk in
their Lemma 1 and we derive the parameter norm in the same way:

$$\mathbb{E}_\epsilon\left[R(\hat{\theta})\right] \to \sigma_\epsilon^2 + \begin{cases} 0 + \sigma_\epsilon^2 \frac{\gamma}{1-\gamma} & \text{for} \quad \gamma < 1 \\ \beta^\top\beta\left(1 - \frac{1}{\gamma}\right) + \sigma_\epsilon^2 \frac{1}{\gamma-1} & \text{for} \quad \gamma > 1 \end{cases}, \tag{92}$$

$$\mathbb{E}_\epsilon\left[\|\hat{\theta}\|_2^2\right] \to \begin{cases} \beta^\top\beta + \sigma_\epsilon^2 \frac{\gamma}{1-\gamma} & \text{for} \quad \gamma < 1 \\ \beta^\top\beta\frac{1}{\gamma} + \sigma_\epsilon^2 \frac{1}{\gamma-1} & \text{for} \quad \gamma > 1 \end{cases}, \tag{93}$$

We see in Figure 9 that the theory form Hastie et al. (2022) for direct regression, the numerical
results for direct regression and our PCA-regression results without compression ($\hat{d} = m$) match
and therefore further support our theory.

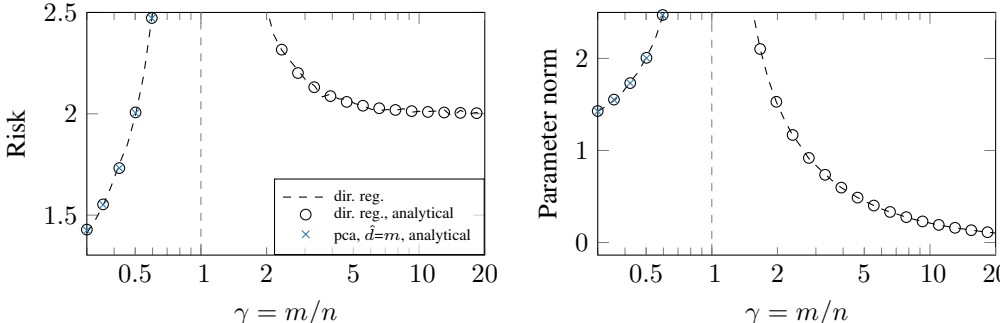

Figure 9: **Supervised results on isotropic data: analysis.** We compare analytical solutions from
Theorem 1 for the PCA-regression without compression ($\hat{d} = m$) with 1) analytical solution for
direct regression and 2) simulations for direct regression

Complementary to Figure 2, we can decompose the risk and parameter norm according to Theorem 1
in a bias and variance term. The results for this decomposition are shown in Figure 10. We can see
that the bias term is nonzero for all $\gamma$ and increases for larger $\gamma$. Further, we observe a decrease of
the variance term. In contrast, in the direct regression model, the variance term reaches a peak at
$\gamma = 1$ and therefore forms the classical bias-variance decomposition trade-off for $\gamma < 1$.

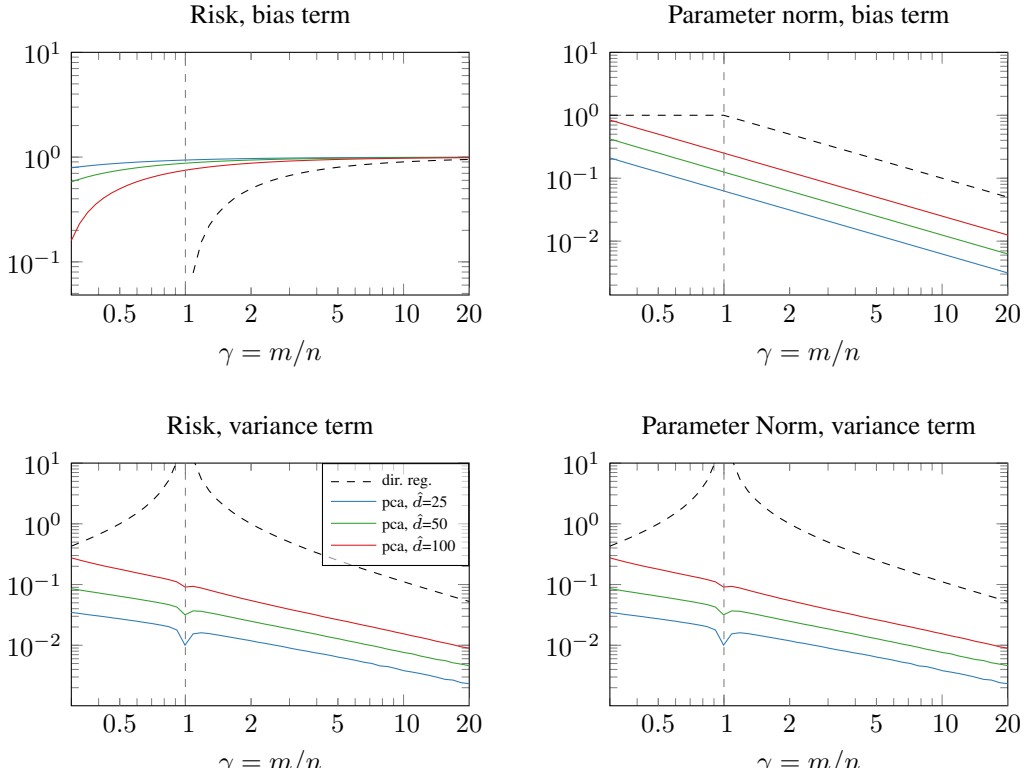

Figure 10: **Bias-variance decomposition of supervised results on isotropic data: analysis.** Same as Figure 2 but decomposed in the bias and variance terms of Theorem 1. In all plots the direct regression results are given as comparison. *Left:* Risk. *Right:* Parameter norm. *Top:* Bias terms. *Bottom:* Variance terms.

## E.2 LATENT VARIABLE DATA

**Complementary result for parameter norm**  Complementary to the results of the risk for $\alpha = 0$ and $\alpha = 0.25$ in Figure 3, we show the results for the parameter norm in Figure 11. We observe that similarly to the isotropic case, the parameter norm decreases monotonically for larger $\gamma$. This indicates simpler solution for larger $\gamma$ also in the latent variable data case.

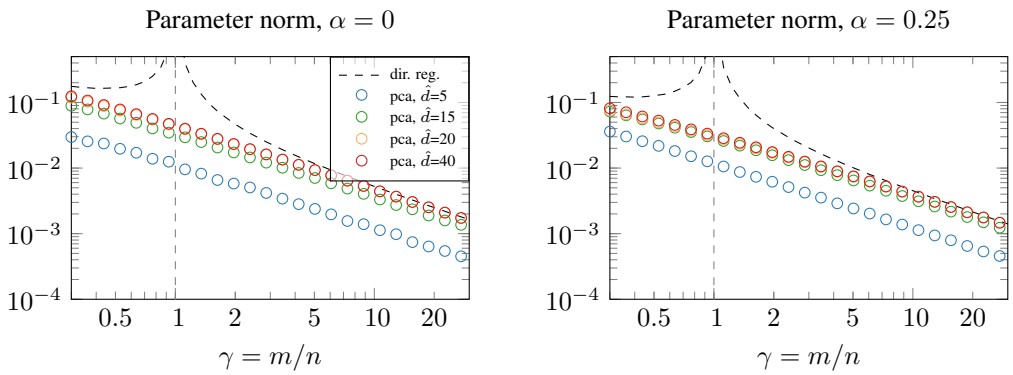

Figure 11: **Supervised parameter norm on latent variable data: simulation.** *Left:* Parameter norm of models for data generated with $\alpha = 0$. *Right:* Equivalent results with $\alpha = 0.25$. This figure complements Figure 3

**Varying noise levels**   In Figure 12 we show the results for risk and parameter norm for different values of $\sigma_y^2$. In the main text we only present results for $\sigma_y^2 = 0$ but the results here show that our conclusion hold also for additive output noise. Of course, the associated risk increases with the noise level but interestingly, the found solution as the same parameter norm.

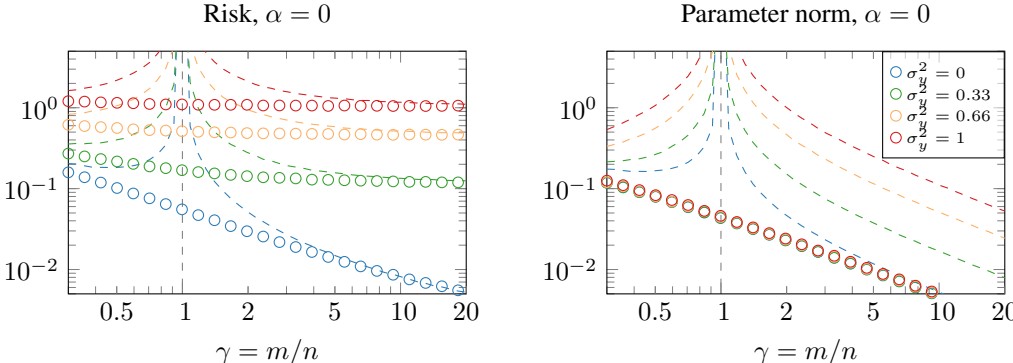

Figure 12: **Supervised results for different $\sigma_y^2$: simulation.** We have the same experimental setup as in Figure 3 with $\alpha = 0$ but vary the amount of output noise. Full lines show the PCA-model with $\hat{d} = d$; dashed lines show the direct regression model for the specific noise level.

**Details of phenomenon for $\alpha > 0$**   In Figure 3 one could observe for $\alpha > 0$ that for a certain range of $\gamma > 1$, the optimal PCA-regression model is worse than the direct regression model. We investigated the details of this observation in the following ways, which we visualized in Figure 13:

1. We increased the range of $\gamma$ for higher values. Doing so, we can observe that results for direct regression (dashed line) and the PCA-regression model ($\times$-marks) converge again.

2. We conduct more experiments with a wider range of values for $\alpha$. In all tested values, the same phenomenon is visible.

3. Analysing the uncertainty of our risk estimates over the 200 averaged simulations, we note that the difference lies within the standard deviation of our risk estimates. This originates from the limited number of test samples (=400) to estimate the risk.

Therefore, we conclude that this phenomenon is an artifact of our experimental setup.

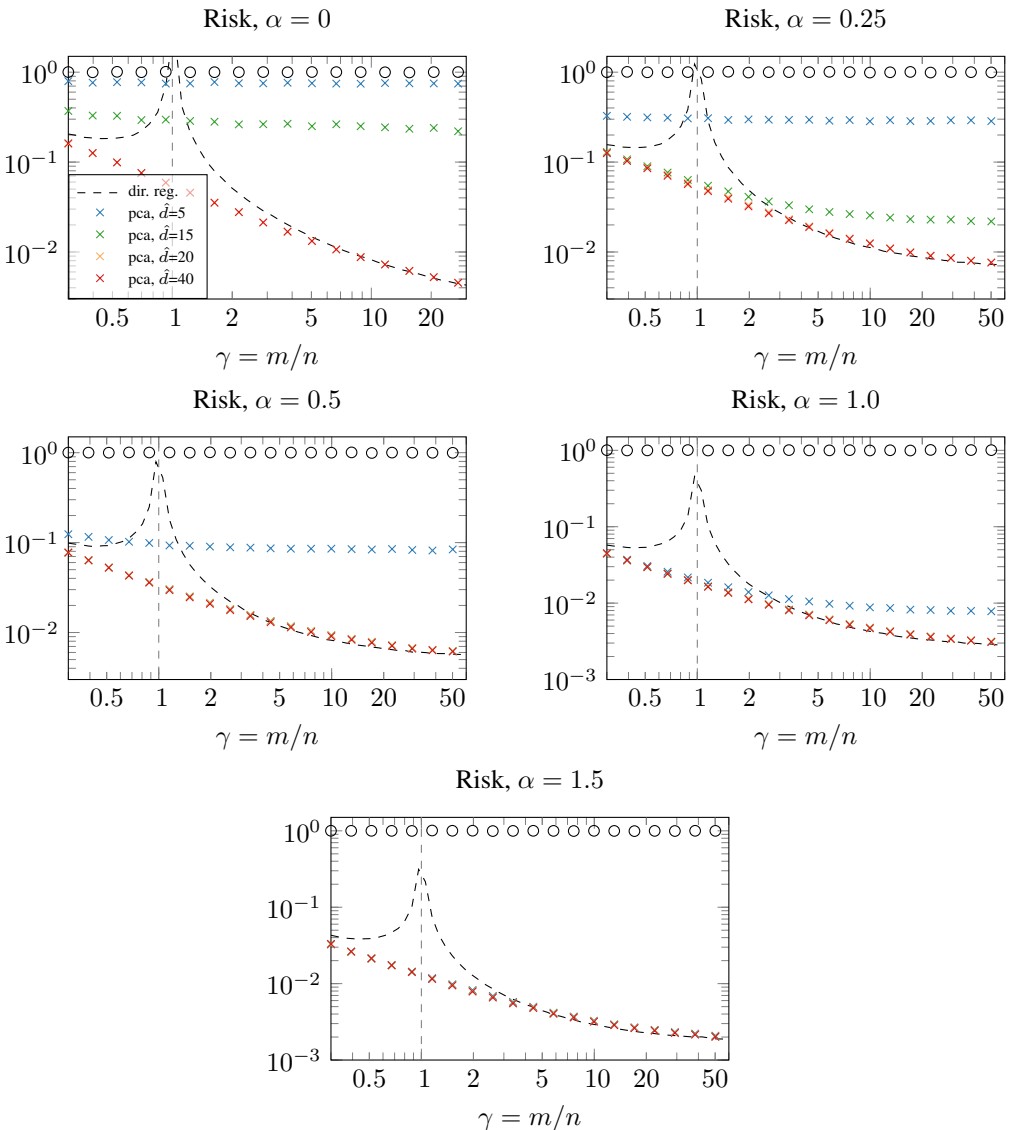

Figure 13: **Supervised risk on latent variable data: simulation.** Top row is a repetition of Figure 3. Remaining rows are for higher values of $\alpha = \{0.5, 1, 1.5\}$.

### E.3 CONNECTION TO RIDGE REGRESSION

In the main paper, we focused on the unregularized linear regression problem. In this part we compare the PCA-regression model with the regularized Ridge regression solution and $\lambda$ as the Ridge parameter

$$\hat{\boldsymbol{\theta}} = (\boldsymbol{X}^\top \boldsymbol{X} + \lambda \boldsymbol{I}_m)^+ \boldsymbol{X}^\top \boldsymbol{y}. \tag{94}$$

In the first part, we look at isotropic data where we have analytical solution. In the second part we compare numerical simulation for the latent variable data generator.

#### E.3.1 ISOTROPIC DATA

For isotropic data we can compare the results from Theorem 1 for the asymptotic risk in the PCA-regression model with Ridge regression. Corollary 6 in Hastie et al. (2022) provides asymptotic results of Ridge regression for isotropic data. For completeness we state the asymptotic Ridge

result:

$$\mathbb{E}_\epsilon\left[R(\hat{\boldsymbol{\theta}}_\lambda)\right] \to \boldsymbol{\beta}^\top\boldsymbol{\beta}\lambda^2 m'(-\lambda) + \sigma_\epsilon^2\gamma\left(m(-\lambda) - \lambda m'(-\lambda)\right) + \sigma_\epsilon^2, \tag{95}$$

with $m(z) = \frac{1-\gamma-z-\sqrt{(1-\gamma-z)^2-4\gamma z}}{2\gamma z}$ and $m'(z)$ as the derivative w.r.t $z$. The optimal Ridge regularization is achieved at $\lambda^* = \sigma_\epsilon^2\gamma/\boldsymbol{\beta}^\top\boldsymbol{\beta}$ which then yields the optimal risk

$$\mathbb{E}_\epsilon\left[R(\hat{\boldsymbol{\theta}}_{\lambda^*})\right] \to \sigma_\epsilon^2\gamma m(-\lambda^*) + \sigma_\epsilon^2. \tag{96}$$

Note that the optimal Ridge regularization strength is a function of $\gamma$ and monotonically increases with $\gamma$. Optimal regularization is not achieved by a single regularization value.

Figure 14 visualizes the comparison of the analytical solutions. The lowest risk for all $\gamma$ is obtained for the optimal Ridge regression solution. Comparing the solutions from Ridge regularization with different $\lambda$ with the solution from PCA-regression with different choices of $\hat{d}$ shows qualitative different behavior for isotropic data. While Ridge regression smoothens out the interpolation peak of direct regression with well tuned $\lambda$, for PCA-regression we require a sufficiently large amount of eigenvectors, i.e. large $\hat{d}$ to obtain a risk lower than the null risk. Overall, optimally tunes Ridge regression outperforms PCA-regression for all $\gamma$.

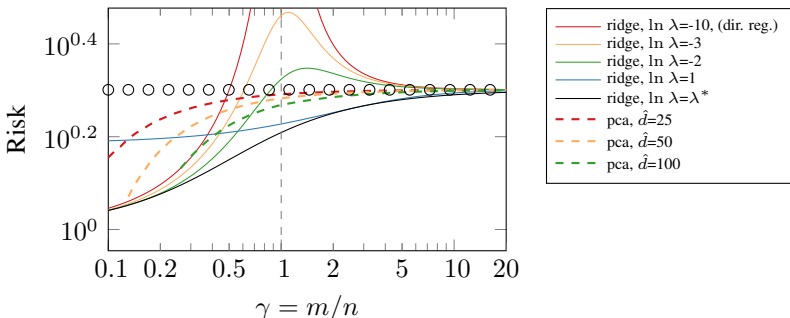

Figure 14: **PCA-regression comparison with ridge regression for isotropic data: analysis.** Solid lines depict the Ridge regularized models. Note that the red solid line with very low $\lambda$ is equal to the unregularized direct regression. Dashed lines depict the analytical PCA-regression model results.

### E.3.2 LATENT VARIABLE DATA GENERATOR

In this setting, we use the latent variable data generator without eigenvalue decay ($\alpha = 0$) and compare our PCA-regression model without model misspecifications, i.e. $\hat{d} = d$, to solutions using different Ridge parameters. We rely on numerical solutions in this section. The results are in Figure 15. While for isotropic data, the comparison between both models shows qualitative different behavior, here for the latent variable data the results indicate a clear connection between both models for large values of $\lambda$. This observation is theoretically justified from the known relationship of both methods on the eigenvalues. Ridge regression lifts all eigenvalues $\boldsymbol{S}$ of the features by a value of $\lambda$

$$\boldsymbol{X}^\top\boldsymbol{X} + \lambda\boldsymbol{I}_m = \boldsymbol{V}^\top(\boldsymbol{S} + \lambda\boldsymbol{I}_d)\boldsymbol{V}. \tag{97}$$

Here, $\boldsymbol{V}$ are the true, non-truncated eigenvectors. In contrast, the PCA-regression model cuts the eigenvalues off at a threshold chosen by $\hat{d}$, which is clear in (7). The main difference is that with ridge regression, there is a smooth change of the risk controlled by the ridge parameter whereas with PCA-regression there is a hard cut-off.

Figure 15 may imply that the optimal Ridge penalty is at $\lambda \to \infty$ as it avoids the interpolation peak at no additional cost. Previous studies have concluded that finite Ridge regularization is better. Gerace et al. (2020) uses the hidden manifold model from Goldt et al. (2020) and Mei & Montanari (2022) use random features model by Rahimi & Recht (2007) which can be seen as a two-layer neural network. Both studies conclude that finite $\lambda$ achieves optimal regularization. However, we are working with linearly separable data, which is closer to the latent space model in Hastie et al.

(2022). The difference to our setup is, that for us both, the data generating process and the PCA-regression model have a low-dimensional embedding. The conclusion in Hastie et al. (2022) that the best Ridge regularization is $\lambda = 0$ and achieved in $\gamma > 1$ may hold for us as well but is difficult to proof with our numerical results in Figure 15.

Optimal penalty at $\lambda \to \infty$ for Ridge regularized problems was also observed in previous studies with different setups to ours. Both Mignacco et al. (2020) and Loureiro et al. (2021) study the classification of high-dimensional (isotropic) Gaussian-mixtures of balanced data from each mixture and show that large $\lambda$ are necessary to reach the Bayes-optimal performance. Thrampoulidis et al. (2020) studies a similar model for the classification of Gaussian mixtures as well as for a multinomial logit model where they identify that the class averaging algorithm, which is equal to Ridge regression with $\lambda = \infty$, performs optimal in some settings.

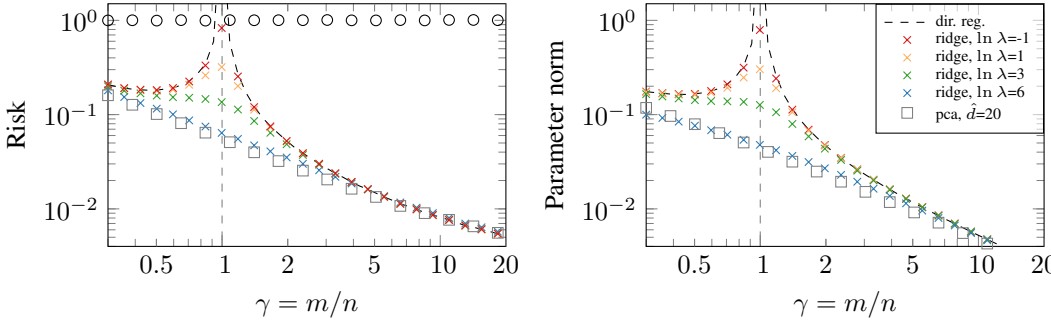

Figure 15: **PCA-regression comparison with ridge regression for latent variable data: simulation.** We highlight the similarity of the results obtained with large ridge parameter to our PCA-regression model. *Left:* Risk. *Right:* Parameter norm.

### E.4   REAL-WORLD EXAMPLE: GENETICS

**Background**   The Diverse MAGIC Wheat data set[1] is based on 16 founding wheat varieties which were listed between 1935 and 2004. These varieties were interbred to obtain new wheat varieties. From the resulting wheat types the genome of total of 502 wheats were sequenced. This genome sequence consists of $\approx 1.1$ M single nucleotide polymorphisms. Furthermore, phenotypes of the 502 wheat types were analysed, see Scott et al. (2021).

**Data processing**   The genotypes consist of binary features. The binary variables represent equality or difference to a reference genotype. The phenotypes are real-values variables. We choose the phenotype column named 'HET_2' in this example. Missing values for both, genotype and phenotype are replaced with the mean value of the variable. We select a subset of genotypes as inputs randomly at uniform to obtain the necessary $m$ features. Then, we normalize both, genotype and phenotype by z-transformation.

**Data analysis**   In Figure 16 we plot the eigenvalue distribution for the Diverse MAGIC Wheat data set, similar to the ones in Appendix B. We observe that the eigenvalue distribution is heavy tailed. It does not depict a clear example of a low dimensional latent manifold. Therefore, using the PCA-regression model will discard some useful information similar to the isotropic case.

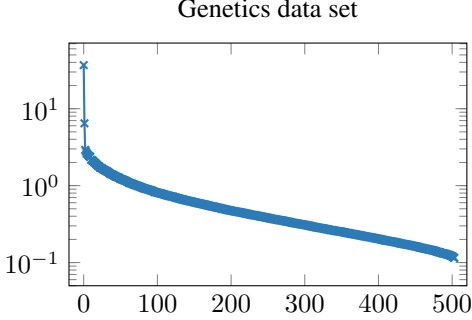

Genetics data set

Figure 16: **Eigenvalue distribution of the Diverse MAGIC Wheat genetics data set.**

---

[1] http://mtweb.cs.ucl.ac.uk/mus/www/MAGICdiverse/

# F PROOFS FOR THE CASE WITH PRE-TRAINING

This appendix derives and proofs the results of section 5.2. First, we derive the results for the estimation of the eigenvectors with the PCA loss in F.1. Then, we show derive the concentration inequality for the PCA loss in F.2.

## F.1 ESTIMATION OF EIGENVECTORS

Here, we prove Lemma 2 for the loss difference for the projection into the true or estimated latent space from (18). We will look at both losses induced by the two projections separately,

*Proof.* First, define the loss for the projection into the true latent space as in (16)

$$\mathcal{L}(\boldsymbol{D}) = \mathbb{E}\left[\|\boldsymbol{x}\|_2^2 - \|\boldsymbol{D}^+\boldsymbol{x}\|_2^2\right]. \tag{98}$$

We can write the second term as the following where we use the cyclic property of the trace on scalars and apply the expectation on $\boldsymbol{x}\boldsymbol{x}^\top$

$$\mathbb{E}\left[\|\boldsymbol{D}^+\boldsymbol{x}\|_2^2\right] = \mathbb{E}\left[\boldsymbol{x}^\top(\boldsymbol{D}^+)^\top\boldsymbol{D}^+\boldsymbol{x}\right] \tag{99}$$

$$= \mathbb{E}\left[\mathrm{Tr}(\boldsymbol{x}^\top(\boldsymbol{D}^+)^\top\boldsymbol{D}^+\boldsymbol{x})\right] \tag{100}$$

$$= \mathrm{Tr}(\boldsymbol{D}^+\boldsymbol{V}\boldsymbol{S}\boldsymbol{V}^\top(\boldsymbol{D}^+)^\top) \tag{101}$$

Since $\boldsymbol{V} = [\boldsymbol{D} \quad \boldsymbol{D}_\perp]$ defined in (24), we obtain

$$\mathbb{E}\left[\|\boldsymbol{D}^+\boldsymbol{x}\|_2^2\right] = \mathrm{Tr}\left([\boldsymbol{I}_d \quad \boldsymbol{0}_{d\times m-m}]\,\boldsymbol{S}\begin{bmatrix}\boldsymbol{I}_d\\\boldsymbol{0}_{m-m\times d}\end{bmatrix}\right) \tag{102}$$

$$= \mathrm{Tr}(\boldsymbol{\Lambda}) = \sum_{i=1}^d s_i \tag{103}$$

Hence, the loss $\mathcal{L}(\boldsymbol{D})$ becomes

$$\mathcal{L}(\boldsymbol{D}) = \sum_{i=1}^m s_i - \sum_{i=1}^d s_i = \sum_{i=d+1}^m s_i = \sum_{i=d+1}^m s_i \tag{104}$$

Second, define the loss for the projection into the estimated latent space as in (17)

$$\mathcal{L}(\hat{\boldsymbol{H}}) = \mathbb{E}\left[\|\boldsymbol{x}\|_2^2 - \|\hat{\boldsymbol{H}}^\top\boldsymbol{x}\|_2^2\right]. \tag{105}$$

Again, we can write the second term as the following by using the same arguments as for the first term

$$\mathbb{E}\left[\|\hat{\boldsymbol{H}}\boldsymbol{x}\|_2^2\right] = \mathrm{Tr}(\hat{\boldsymbol{H}}^\top\boldsymbol{V}\boldsymbol{S}\boldsymbol{V}^\top\hat{\boldsymbol{H}}) = \sum_{i=1}^{\hat{d}}\sum_{j=1}^m (\hat{\boldsymbol{h}}_i^\top\boldsymbol{v}_j)^2 s_j. \tag{106}$$

The last equality holds by switching to vector notation where factors can be combined to squared terms. Hence, the loss $\mathcal{L}(\hat{\boldsymbol{H}})$ becomes

$$\mathcal{L}(\hat{\boldsymbol{H}}) = \sum_{i=1}^m s_i - \sum_{i=1}^{\hat{d}}\sum_{j=1}^m (\hat{\boldsymbol{h}}_i^\top\boldsymbol{v}_j)^2 s_j. \tag{107}$$

Combining both solutions, the loss difference yields

$$\mathcal{L}(\hat{\boldsymbol{H}}) - \mathcal{L}(\boldsymbol{D}) = \sum_{i=d+1}^m s_i - \sum_{i=1}^m s_i + \sum_{i=1}^{\hat{d}}\sum_{j=1}^m (\hat{\boldsymbol{h}}_i^\top\boldsymbol{v}_j)^2 s_j \tag{108}$$

$$= \sum_{i=1}^d s_i + \sum_{i=1}^{\hat{d}}\sum_{j=1}^m (\hat{\boldsymbol{h}}_i^\top\boldsymbol{v}_j)^2 s_j \tag{109}$$

We can multiply the first term by $1 = \|\boldsymbol{v}_j\|_2^2 = \boldsymbol{v}_j^\top \boldsymbol{I} \boldsymbol{v}_j = \boldsymbol{v}_j^\top \hat{\boldsymbol{h}}_i \hat{\boldsymbol{h}}_i^\top \boldsymbol{v}_j = \|\hat{\boldsymbol{h}}_i^\top \boldsymbol{v}_j\|_2^2$,

$$\mathcal{L}(\hat{\boldsymbol{H}}) - \mathcal{L}(\boldsymbol{D}) = \sum_{i=1}^{d} (\hat{\boldsymbol{h}}_i \boldsymbol{v}_j)^2 s_i + \sum_{i=1}^{\hat{d}} \sum_{j=1}^{m} (\hat{\boldsymbol{h}}_i^\top \boldsymbol{v}_j)^2 s_j \tag{110}$$

Now we can combine terms but have to be careful due to different end indices of the sum

$$\mathcal{L}(\hat{\boldsymbol{H}}) - \mathcal{L}(\boldsymbol{D}) = \begin{cases} \sum_{i=1}^{\hat{d}} \sum_{j=1}^{m} (\hat{\boldsymbol{h}}_i^\top \boldsymbol{v}_j)^2 (s_i - s_j) + \sum_{i=\hat{d}}^{d} s_i & \text{for} \quad \hat{d} < d \\ \sum_{i=1}^{\hat{d}} \sum_{j=1}^{m} (\hat{\boldsymbol{h}}_i^\top \boldsymbol{v}_j)^2 (s_i - s_j) & \text{for} \quad \hat{d} = d \\ \sum_{i=1}^{d} \sum_{j=1}^{m} (\hat{\boldsymbol{h}}_i^\top \boldsymbol{v}_j)^2 (s_i - s_j) + \sum_{i=d}^{\hat{d}} \sum_{j=1}^{m} (\hat{\boldsymbol{h}}_i^\top \boldsymbol{v}_j)^2 s_j & \text{for} \quad \hat{d} > d \end{cases} \tag{111}$$

which we can combine into

$$\mathcal{L}(\hat{\boldsymbol{H}}) - \mathcal{L}(\boldsymbol{D}) = \sum_{i=1}^{\min(d,\hat{d})} \sum_{j=1}^{m} (\hat{\boldsymbol{h}}_i^\top \boldsymbol{v}_j)^2 (s_i - s_j) + \underbrace{\sum_{i=\hat{d}}^{d} s_i}_{=0 \text{ for } \hat{d} \geq d} + \underbrace{\sum_{i=d}^{\hat{d}} \sum_{j=1}^{m} (\hat{\boldsymbol{h}}_i^\top \boldsymbol{v}_j)^2 s_j}_{=0 \text{ for } \hat{d} \leq d}. \tag{112}$$

which concludes the proof. $\qquad\square$

### F.2 CONCENTRATION INEQUALITY

Here, we prove the concentration inequality presented in Theorem 2.

*Proof.* Write the loss difference (18) with the same argument as in its derivation by including the factor $\|\boldsymbol{v}_j\|_2^2 = 1$ to the second summation yields

$$\mathcal{L}(\hat{\boldsymbol{H}}) - \mathcal{L}(\boldsymbol{D}) = \sum_{i=1}^{\min(d,\hat{d})} \sum_{j=1}^{m} (\hat{\boldsymbol{h}}_i^\top \boldsymbol{v}_j)^2 (s_i - s_j) + \sum_{i=\hat{d}}^{d} \sum_{j=1}^{m} (\hat{\boldsymbol{h}}_i^\top \boldsymbol{v}_j)^2 s_i + \sum_{i=d}^{\hat{d}} \sum_{j=1}^{m} (\hat{\boldsymbol{h}}_i^\top \boldsymbol{v}_j)^2 s_j \tag{113}$$

Notice that $s_i - s_j \geq 0$ for $j > i$ and $s_i - s_j \leq 0$ for $j < i$. Therefore, we can upper bound it by removing the negative indices from the first summation

$$\mathcal{L}(\hat{\boldsymbol{H}}) - \mathcal{L}(\boldsymbol{D}) \leq \sum_{i=1}^{\min(d,\hat{d})} \sum_{j>i}^{m} (\hat{\boldsymbol{h}}_i^\top \boldsymbol{v}_j)^2 |s_i - s_j| + \sum_{i=\hat{d}}^{d} \sum_{j=1}^{m} (\hat{\boldsymbol{h}}_i^\top \boldsymbol{v}_j)^2 s_i + \sum_{i=d}^{\hat{d}} \sum_{j=1}^{m} (\hat{\boldsymbol{h}}_i^\top \boldsymbol{v}_j)^2 s_j \tag{114}$$

We simplify by denoting the three terms as

$$\mathcal{L}(\hat{\boldsymbol{H}}) - \mathcal{L}(\boldsymbol{D}) = a + b + c \tag{115}$$

Now we define the probability that this upper bound on the loss difference is larger than a chosen real $t$. We can upper bound this expression by applying the union bound

$$P(a + b + c > t) \leq P(a > t) + P(b > t) + P(c > t) \tag{116}$$

Recall Corollary 4.1 from Loukas (2017): We have that for any weights $w_{ij}$ and real $t > 0$ that

$$P\left( \sum_{i \neq j} w_{ij} (\hat{\boldsymbol{h}}_i^\top \boldsymbol{v}_j)^2 > t \right) \leq \sum_{i \neq j} \frac{4 w_{ij} k_j^2}{n_p t (s_i - s_j)^2} \tag{117}$$

where $k_j^2 = \mathbb{E}\left[ \|\boldsymbol{x}\boldsymbol{x}^\top \boldsymbol{v}_j\|_2^2 \right] - s_j^2$ and $w_{ij} \neq 0$ when $s_i \neq s_j$ and $sgn(s_i - s_j)2s_i > sgn(s_i - s_j)(s_i + s_j)$.

In accordance with this Corollary, we define

$$w_{ij} = \begin{cases} |s_i - s_j| & \text{for} \quad i \leq \min(d, \hat{d}), j > i \\ s_i & \text{for} \quad \hat{d} \leq i \leq d, \forall j \\ s_j & \text{for} \quad d \leq i \leq \hat{d}, \forall j \\ 0 & \text{otherwise} \end{cases} \tag{118}$$

Hence, we obtain

$$P\left(\mathcal{L}(\hat{\boldsymbol{H}}) - \mathcal{L}(\boldsymbol{D}) > t\right) \leq$$

$$\leq \frac{4}{t\,n_p} \left( \sum_{i=1}^{\min(d,\hat{d})} \sum_{j=i+1}^{m} \frac{k_j^2}{|s_i - s_j|} + \sum_{i=\hat{d}}^{d} \sum_{j=1}^{m} \frac{k_j^2 s_i}{(s_i - s_j)^2} + \sum_{i=d}^{\hat{d}} \sum_{j=1}^{m} \frac{k_j^2 s_j}{(s_i - s_j)^2} \right), \quad (119)$$

with $k_j^2 = s_j(s_j + \mathrm{Tr}(\boldsymbol{C}))$ from Corollary 4.3 in Loukas (2017). This Corollary holds for our latent variable data generator. This concludes the proof. □

## G ADDITIONAL NUMERICAL RESULTS FOR THE CASE WITH PRE-TRAINING

While in Section 5.3 we concentrate on the well specified case with $\hat{d} = d$, here we show the effect of model misspecification. Specifically for the same data generator with $d = 20$ we choose $\hat{d} = \{15, 20, 40\}$. The results for the full risk over all $\mu$ is in Figure 17 and slices of this figure are in Figure 18. We can observe a qualitatively similar behavior to model misspecification as when we train fully supervised, see Section 4.2 or Figure 3: For $\hat{d} < d$, the risk is high and does not decrease significantly for larger $\gamma$. For $\hat{d} \geq d$, the risk decreases as expected. Therefore, from our observations the conclusions for model misspecification from the supervised case translates to the case with pre-training.

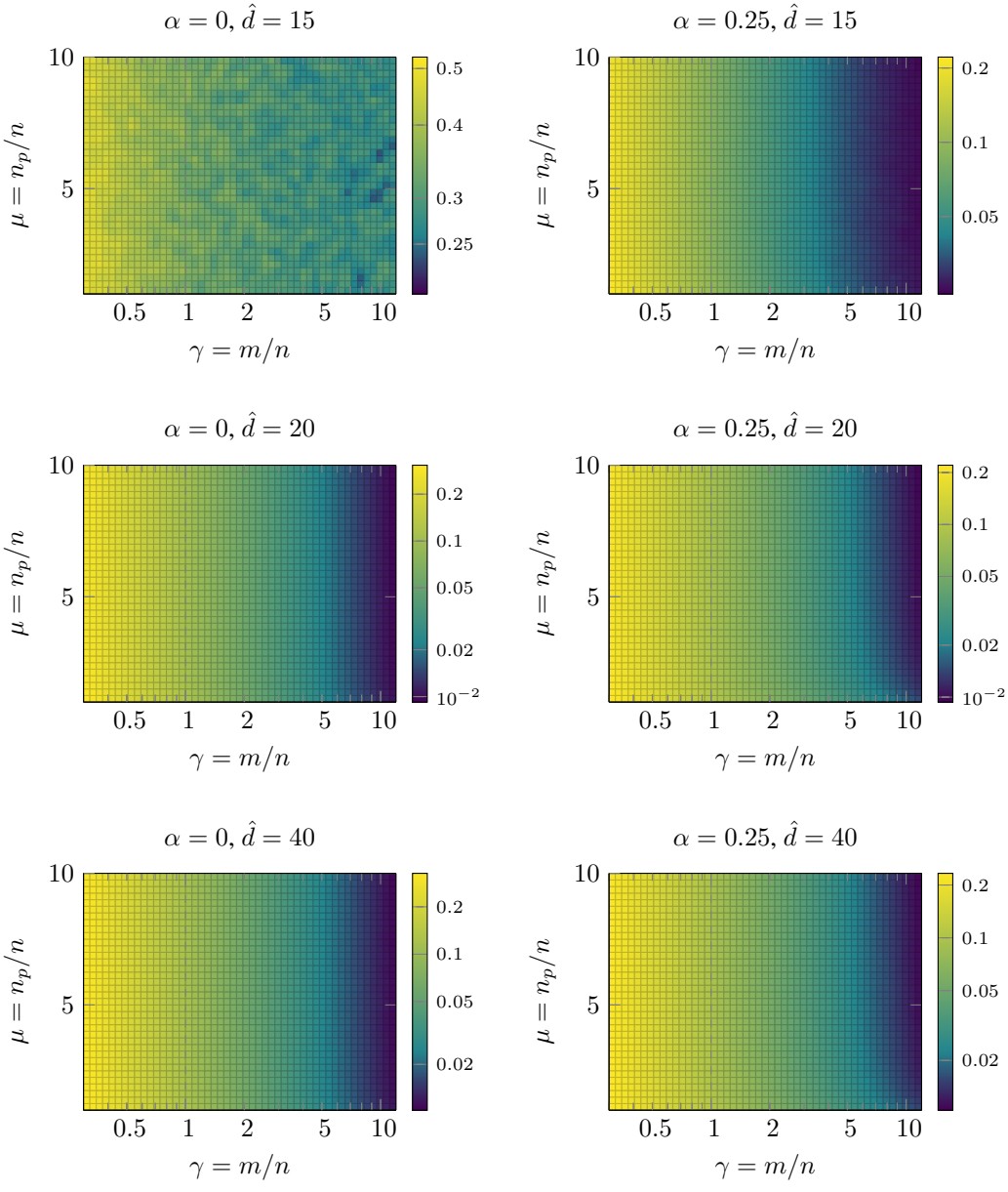

Figure 17: **Pre-training risk on latent variable data: simulation.** We use the latent variable data generator with $d = 20$. In this simulations we show the effect of model misspecification. *Left:* Risk for data with feature covariance eigenvalue decay of $\alpha = 0$. *Right:* Same setup but for $\alpha = 0.25$. *Top row: $\hat{d} < d$. Middle row: $\hat{d} = d$.* This is a repetition of Figure 5. *Bottom row: $\hat{d} > d$.*

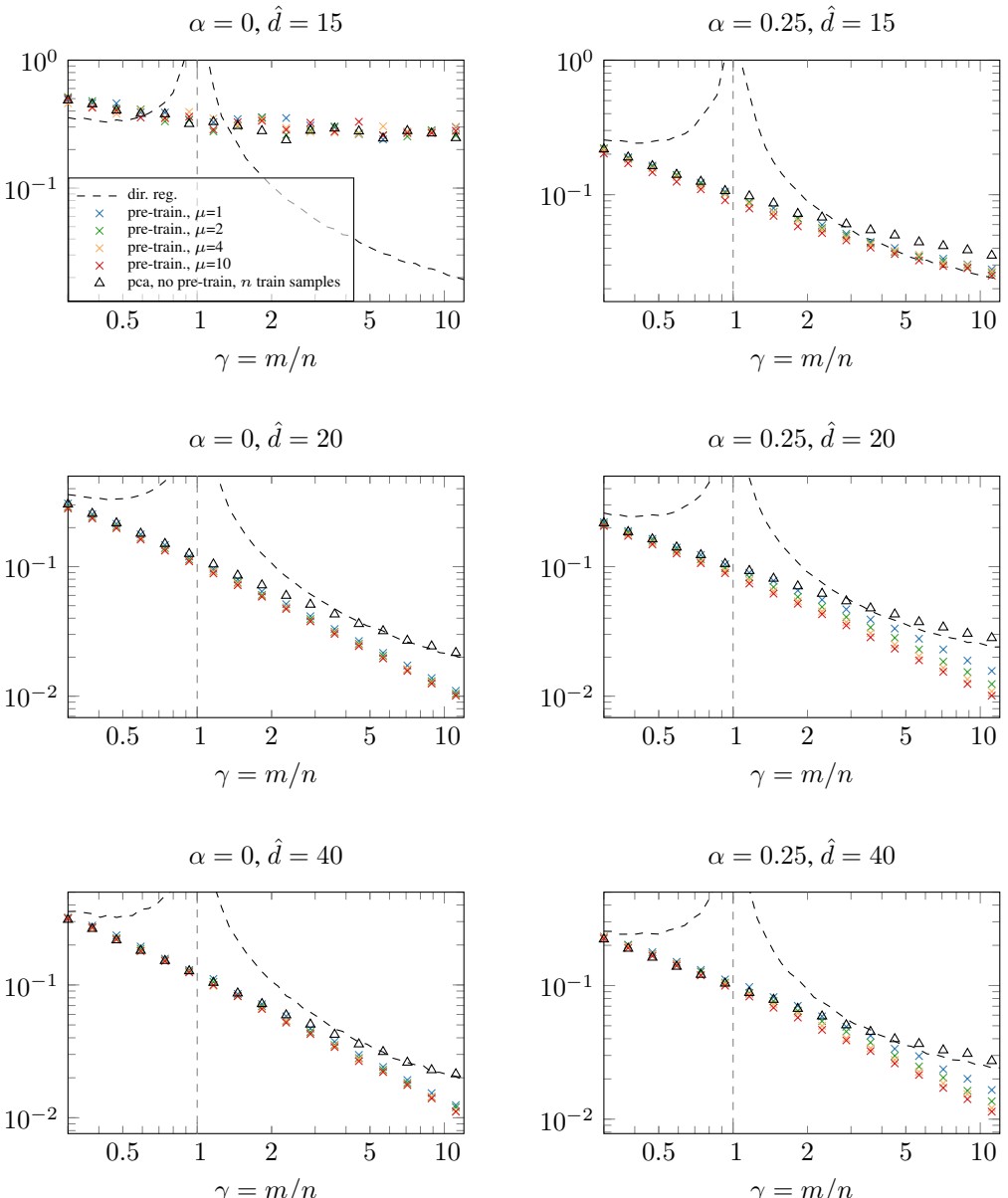

Figure 18: **Pre-training risk for different horizontal slices of Figure 17: simulation.** *Left:* Risk for data with feature covariance eigenvalue decay of $\alpha = 0$. *Right:* Same setup but for $\alpha = 0.25$. *Top row:* $\hat{d} < d$. *Middle row:* $\hat{d} = d$. *Bottom row:* $\hat{d} > d$. The middle right figure is a repetition of Figure 6.

