# OpenReview forum: "No Double Descent in PCA: Training and Pre-Training in High Dimensions"
_ICLR.cc/2023/Conference — Submitted to ICLR 2023_

### Official Review · Reviewer_eejS · 2022-10-15

**Confidence:** 3
**Correctness:** 3
**Technical Novelty And Significance:** 3
**Empirical Novelty And Significance:** 2
**Recommendation:** 3

**Clarity, Quality, Novelty And Reproducibility:**

I think each step of the paper is pretty clearly written, but, as noted above, I don't totally understand the overall point of the paper. I think the authors have made paper's results are straightforward to reproduce.

**Strength And Weaknesses:**

The results and experiments in the paper are individually interesting, but I'm not totally clear on how they come together to tell a cohesive story about overparameterization in encoder-decoder models.

1. What is the purpose of Figure 4 and the discussion of ridge regression? I don't think these are new results (e.g. they seem to be covered in Hastie et al. 2022), and it's not clear to me how they contribute to the discussion of PCA.
2. Section 5.2 is supposed to be a theoretical analysis of the pre-trained PCA model. But the main result of this section (Theorem 2) is just about estimating covariance matrices from data. This definitely seems like a critical step along the way to understanding the risk of this pre-trained model, but this only gets us part of the way there. There is some discussion below Theorem 2 about connections to the risk, but these results seem like high-level qualitative discussions, rather than theory.
3. I don't immediately see how the paper's results relate to PCA-followed-by-regression in practice or other machine learning models: (3i) all of the results are about a fixed $\hat d$ in the paper. But in practice, one has to select $\hat d$ somehow, and this selection will change the risk; it's hard to say if the title of the paper ("No double descent in PCA") will hold in this realistic case. (3ii) There's not any discussion about how the models the authors study relate to actual models in practice. E.g. a discussion along the lines of Section 1.2 of Hastie et al. 2022 would be really helpful, especially since I don't think the model studied here is actually used in practice (see point (3i)).
4. In the conclusion, I'm not sure I see why real datasets would present such a challenge here. Could one not use a held-out set to estimate the risk and use subsampling to vary $\gamma$?

I also have a few smaller points about the paper's experiments
1. Some conclusions are drawn about the relative order of the lines in on the right of Figure 3. I don't think these are valid conclusions from this plot. The differences seem to be on the order of about 0.01 or less, and (I think) the risk is estimated here by averaging over 400 test points. This doesn't seem like enough test points to make accurate inferences about differences this small.
2. In Section 5.3, $\hat d = d$ is used because "the effects of misspecified models is elaborated in Section 4.2." But in Section 4.2, we are studying a different model. Why should we expect the results of that section to relate to the pre-training regime studied here?
3. "We see that the loss with pre-training is lower, which indicates that using different data sets and therefore more diverse data is advantageous." I don't think you can conclude this. Pre-training is giving more data overall. You would have to compare to both methods getting the same volume of data to make a conclusion like this; otherwise, I think a reasonable explanation for the behavior we are seeing here is that pre-training is getting more data.


Misc smaller points
- "The variance (second) term is controlled..." and "The first term represents the variance and the last one the bias." Both of these statements are proceeded by two equations; which equation are these statements about (or is it both)?
- Lemma 1 is about centered features, but I don't think centered features were previously discussed.
- "We compute risk and parameter norm as defined in Lemma 1". Lemma 1 has two equations for risk / norm (one the actual risk and the other the expectation). Which one is being used here?

**Summary Of The Paper:**

This is a theoretical paper studying a method for regression in high dimensions. In particular, it studies the use of PCA followed by linear regression. Their motivation for doing so is to better understand overparameterization in more general encoder-decoder models. The authors provide an explicit formula for the expected risk and prove the asymptotic behavior of this risk under a particular random covariate model. Empirically, these formulas do not show the "double descent" behavior that comes up in many other overparameterized models. The authors further study the effect of pre-training by considering a second regression method. They assume access to a (possibly large) pool of covariates $x_i$ without responses $y_i$. This large pool of covariates is used to estimate the top-$\hat d$ eigenvectors of the distribution of the $x_i$; then, given a training $\{(x_n, y_n)\}$, regression is done in the space of these estimated top-$\hat d$ eigenvectors. The authors prove a concentration inequality about how well these top-$\hat d$ eigenvectors are estimated.

**Summary Of The Review:**

Overall, I think the paper's current results need to either be presented significantly differently or some new results are needed to better connect the current results to the rest of machine learning practice / theory. that said, I'm not an expert in the area of PCA, so I'm open to being convinced that the results are meaningful as-is.

---

> ### Author Response · Authors · 2022-11-14
> **Author response (2/2)**
>
> **References**
>
> [Xu and Hsu, 2019] Ji Xu and Daniel J Hsu. On the number of variables to use in principal component regression. Advances in neural information processing systems, 32, 2019.
>
> [Massy, 1965] William F Massy. Principal components regression in exploratory statistical research. Journal of the American Statistical Association, 60(309):234–256, 1965.
>
> [Wang and Abbott, 2008] Kai Wang and Diana Abbott. A principal components regression approach to multilocus genetic association studies. Genet. Epidemiol., 32(2):108–118, 2008

---

> ### Author Response · Authors · 2022-11-14
> **Author response (1/2)**
>
> We thank the reviewer for the clear review. The concerns that you raised about the main motivation and overall point of the paper, motivated us to highlight this more clearly in the paper. We try to convince you in the following that PCA-regression is a highly used model in real-world applications and our results are therefore meaningful to a larger community. We address your review pointwise.
>
> **Answers to general comments**:
> 1. The connection between our PCA-regression model and Ridge regularization is of interesting nature as the example on data from the latent variable data generator shows (similar behavior for $\lambda\rightarrow\infty$). However, this connection is not the main contribution and was therefore moved to Appendix E3. Still, we elaborated in more detail on this connection by providing additional experiments with analytical solutions for isotropic data. Further, we discuss the observation that $\lambda\rightarrow\infty$ seems to be the best regularization strength in our setup with results from literature and related models, see Appendix E3.2.
> 2. Section 5.2 is a general analysis of pre-training. We updated the paragraph `Connection to risk` to have less of a high-level discuss but rather highlight the concrete connection to previous work. Theorem 2 now fills the missing gap and introduces conditions on when to use the results from [Xu and Hsu, 2019].
> 3. (3i) the selection of the latent dimension $\hat d$ is usually done during model selection---a process which we do not cover in the paper. However, we intensively discuss the effect of model misspecification, i.e. $\hat d \neq d$ on the risk. Therefore, we cover all practical aspects of the risk. (3ii) We would like to correct the reviewer here that the model is indeed used in practice. We already present references of the real-world use of PCA-regression in the beginning of Section 4. These include:
> -[Massy, 1965]: This paper uses the income of families and education level from Chicago in 1950 to regress the ownership rate of televisions, refrigeratros, and similar.
> -[Wang and Abbott, 2008]: The authors develop a new PCA-regression model for genetic association studies to determine genetic variants of complex human diseases. They work with an example of 770,394 features and only 57 individuals.
> - Further, we conducted a PubMed search of the terms "principle component regression" OR "PCA-OLS" OR "Principle component analysis regression", which yields 772 results in the last 20 years (2002-2022). This highlights the heavy use of the PCA-regression model in real-world applications. To address the reviewer's concern we added a new related work section in Appendix A about the application of PCA-regression. Here we include a discussion about the selection of $\hat d$ as well.
> 4. We fully agree with the reviewer here. We now added a real-world dataset from the genetics community as an example for supervised training in section 4.2 and Figure 4. This example highlights key observations for the PCA-regression model from our simulations and therefore a clear connection to real-world examples.
>
> **Answers to points about experiments**:
> 1. We investigated this in more detail and agree with the reviewer that it is an artifact of our experimental setup. We removed the conclusion from the main paper but included our more rigorous experimental analysis (more values of $\alpha$ and larger $\gamma$) in Appendix E4.
> 2. In Section 5, we do study the same model as in section 4 but train the single parts in different stages. The effect of model misspecification $\hat d \neq d$ is therefore the same. We make this more clear with experiments of misspecified models in the case of pre-training which we present in Appendix G. We point to these experiments in the main text. Hence, we justify concentrating on well-specified models in Section 5.3.
> 3. We did observe in a small-scale experiment the same behavior when using $2n$ samples for the model without pre-training and for the model with pre-training $n_p$ samples for pre-training and $n$ samples for training. However, this behavior was not studied in detail and is not the focus of this work. Therefore, we toned down the statement and leave it open for future investigation.
>
> **Answers to misc smaller points**:
> - These statements are about both equations. We state this more clearly now.
> - The necessity of centered features is actually not a necessary condition. We therefore removed this condition.
> - We referred here to the definition in the text of Lemma 2. We state this more clearly now.
>
> We hope that the reviewer agrees with the relevance of the studied PCA-regression model and that it is a model used in practice. Our submission therefore fills gaps for fundamental guarantees of practitioners to use this model.

---

### Official Review · Reviewer_JqQV · 2022-10-21

**Confidence:** 4
**Correctness:** 2
**Technical Novelty And Significance:** 3
**Empirical Novelty And Significance:** 3
**Recommendation:** 5

**Clarity, Quality, Novelty And Reproducibility:**

**Clarity**: The overall flow of the paper is confusing, mostly because of the different data models considered in the different parts which create a discontinuity in the reading. It is also not always clear what plots follow from the theoretical results and what plots are purely numerical. I encourage the authors to be more explicit about that.

**Quality**: Overall, I find this work discusses an interesting problem, highlighting some interesting behaviour in simple models that could be of interest.

**Novelty**: To my best knowledge, exact asymptotics for the combination of PCA + least squares regression is novel. The concentration bound of Theorem 2 strongly builds on previous work [Loukas '17], but to my knowledge the discussion in this context is also original.

**Reproducibility**: The code for reproducing the figures is provided, and the theoretical results are discussed in details.

**Details Of Ethics Concerns:**

This work is of theoretical nature.

**Strength And Weaknesses:**

**Weaknesses**:
The major shortcoming of this work is that the theoretical results required to support the discussion are largely incomplete. As I discuss in the summary, this works deals with 3 data models and 2 estimation models. In the first part of the work, estimation model (1) is discussed in the context of data models (a) & (b), however the asymptotic results in Theorem 1 are only for data model (a), the rest of the plots relying on numerical observations. In the second part of the work, estimation model (2) is discussed in the context of data model (c), a simplified version of model (b) where the noise is orthogonal to the optimal decoder. Here, only a concentration bound on the estimation of the decoder is presented, allowing only for an informal discussion of the generalisation error and a direct comparison with the full supervised setting in the first part. While numerics might seen enough to draw the phenomenology, it has strong limitations regarding finer questions, for instance whether the observation that direct regression can outperform PCA in model (1)+(b) at $\alpha>0$ (see **[Q2]** below.)

A minor weakness is the lack of comparison with real data, which would give an interesting support to the relevance of the models.

**Strengths**:
The overall discussion and conclusions of the paper are interesting, and have many potential follow ups. I believe that having a complete theory (even if heuristic) for at least Part I would make this an interest contribution for the ICLR community.


**Questions**:
- **[Q1]**: In Fig. 1 left, why are the analytical curves noisy? According to Theorem 1, they should be deterministic quantities of the model parameters.

- **[Q2]**: Are the authors confident of the observation in the top of Page 6, that for $\alpha>0$ and certain $\gamma>1$ the performance of regression on the top of PCA is worst than direct regression? Given that these are numerical results with noisy curves and in Fig. 3 (right) the difference is quite small, I believe this is not sufficient to support this claim. Can the authors provide an example (maybe larger $\gamma$ or larger $\alpha$) where this behaviour is pronounced?

- **[Q3]**: In the discussion around Fig. 4, the authors compare PCA to a fixed $\ell_2$ penalty. The figures suggest that the optimal regularisation here is actually $\lambda = \infty$. This is in contrast to other toy settings for studying overparametrisation, e.g. random features model, where the optimal regularization is finite [Mei & Montanari '22; Gerace et al. '20]. Is this observed in other cases (e.g. $\hat{d}\neq d$, $\alpha > 0$, $\sigma_{y}>0$)? Can the authors make sense of that?

As related side note, optimal $\lambda = \infty$ was observed in classification with balanced & isotropic Gaussian mixture data in [Mignacco et al. '20; Thrampoulidis et al. '20; Loureiro et al. '21], where it corresponds to the optimal Bayes-optimal plug-in estimator for this problem. However, for class imbalance [Mignacco et al. '20] and anisotropic mixtures [Loureiro et al. '21] the optimal regularizer is finite. Maybe there is an interesting connection to be drawn.


**Comments**:
- **[C1]**: Despite being employed by different authors, for the isotropic model the terminology over- and under-parametrised is misleading. Since the number of parameters in both the data model and the statistical model are proportional to the features dimension $m$, increasing the number of parameters actually decrease the sample complexity $\gamma^{-1} = n/d$ of the problem, making it harder to learn (as shown in the right-side of the peak in Fig. 2).

- **[C2]**: Although only the $\mu>1$ (less pre-training data than training data) is considered here, the case $\mu<1$ might also be of interest, since it is common in the context of transfer learning, where the estimation of a latent model for a big data set might be publicly available, and the statistician is not able to perform PCA on the combined data (either because she has not access to the pre-train data or because she lacks computational resources). Do the authors expect a different phenomenology in this case?

- **[C3]**: The latent data model studied in this work is asymptotically equivalent to the random features / hidden manifold model studied in [Mei & Montanari '22; Gerace et al. '20]. Indeed, Gaussian equivalence [Mei & Montanari '22; Goldt et al. '22; Hu, Lu '20] asymptotically implies that:
$$
x_{i} = \sigma(D z_{i}) \asymp D z_{i} + \kappa_{\star}\xi_{i}
$$
for some constants $\kappa_{1},\kappa_{\star}$ and an independent noise $\xi\sim\mathcal{N}(0,I)$ (for simplicity assume $\sigma$ odd), which is exactly the latent model in eq. (1). It would be nice if the authors comment on this connection and cite the related literature.


**Smaller typos and suggestions:**
- Transposes $^{\top}$ are missing in a couple of equations, e.g. eqs. (2), (4).
- Just below eq. (7), it would be good to write $\beta\in\mathbb{R}^{m}$ and the transformation of the noise explicitly for the sake of clarity.
- $\hat{y}_{0}$ in the expectation above eq. (9) is not defined. I guess the authors mean an expectation over a new sample $(x,y)$ with $\hat{y} = \hat{y}(x)$?
- In the statement of Theorem 1 for the isotropic case ($d=m$) it would be clearer to keep only one of these two dimensions, or write explicitly $d=m$.
- In figures 3 and 4, which curves are theoretical or numerical results? It would be nice if the authors could be more precise in the captions.
- Not as related to the discussion in the paper, but since the authors mention in the related works. Double descent has discussed in the context of random features classification in [Gerace et al. '20] (before [Wang et al. '21]) and of ensembling methods also in [Ascoli et al. '20; Loureiro et al. '22]. The interpolation peak has been observed in a few precursors to [Belkin et al. '18], including in analytical results for least squares in the isotropic model in [Krogh, Hertz 91'; Opper '95] and numerically for neural networks in [Geman et al. '92].

**References**

[[Gerace et al. '20]](http://proceedings.mlr.press/v119/gerace20a.html): F Gerace, B Loureiro, F Krzakala, M Mezard, L Zdeborova. *Generalisation error in learning with random features and the hidden manifold model*. Proceedings of the 37th International Conference on Machine Learning, PMLR 119:3452-3462, 2020.

[[Mignacco et al. '20]](http://proceedings.mlr.press/v119/mignacco20a.html): F Mignacco, F Krzakala, Y Lu, P Urbani, L Zdeborova. *The Role of Regularization in Classification of High-dimensional Noisy Gaussian Mixture*. Proceedings of the 37th International Conference on Machine Learning, PMLR 119:6874-6883, 2020.

[[Thrampoulidis et al. '20]](https://proceedings.neurips.cc/paper/2020/hash/6547884cea64550284728eb26b0947ef-Abstract.html): C Thrampoulidis, S Oymak, M Soltanolkotabi. *Theoretical Insights Into Multiclass Classification: A High-dimensional Asymptotic View*.  Part of Advances in Neural Information Processing Systems 33 (NeurIPS 2020).

[[Loureiro et al. '21]](https://proceedings.neurips.cc/paper/2021/hash/543e83748234f7cbab21aa0ade66565f-Abstract.html) B Loureiro, G Sicuro, C Gerbelot, A Pacco, F Krzakala, L Zdeborová. *Learning Gaussian Mixtures with Generalized Linear Models: Precise Asymptotics in High-dimensions*. Part of Advances in Neural Information Processing Systems 34 (NeurIPS 2021).

[[Goldt et al. '22]](https://proceedings.mlr.press/v145/goldt22a.html): S Goldt, B Loureiro, G Reeves, F Krzakala, M Mezard, L Zdeborova. *The Gaussian equivalence of generative models for learning with shallow neural networks*. Proceedings of the 2nd Mathematical and Scientific Machine Learning Conference, PMLR 145:426-471, 2022.

[[Hu, Lu '20]](https://arxiv.org/abs/2009.07669): H Hu, YM Lu. *Universality Laws for High-Dimensional Learning with Random Features*. arXiv: 2009.07669 [cs.IT]

[[Ascoli et al. '20]](http://proceedings.mlr.press/v119/d-ascoli20a.html): S D’Ascoli, M Refinetti, G Biroli, F Krzakala. *Double Trouble in Double Descent: Bias and Variance(s) in the Lazy Regime*. Proceedings of the 37th International Conference on Machine Learning, PMLR 119:2280-2290, 2020.

[[Loureiro et al. '22]](https://proceedings.mlr.press/v162/loureiro22a.html): B Loureiro, C Gerbelot, M Refinetti, G Sicuro, F Krzakala. *Fluctuations, Bias, Variance & Ensemble of Learners: Exact Asymptotics for Convex Losses in High-Dimension*. Proceedings of the 39th International Conference on Machine Learning, PMLR 162:14283-14314, 2022.

[[Krogh, Hertz 91']](https://papers.nips.cc/paper/1991/hash/8eefcfdf5990e441f0fb6f3fad709e21-Abstract.html): A. Krogh, J. Hertz. *A Simple Weight Decay Can Improve Generalization*. Advances in Neural Information Processing Systems 4 (NIPS 1991)

[[Geman et al. '92]](https://ieeexplore.ieee.org/document/6797087): S. Geman, E. Bienenstock, R. Doursat. Neural Networks and the Bias/Variance Dilemma. in Neural Computation, vol. 4, no. 1, pp. 1-58, Jan. 1992.

[Opper '95]: M. Opper. *Statistical Mechanics of Learning : Generalization*. In The Handbook of Brain Theory and Neural Networks. pp. 922--925 (1995)

**Summary Of The Paper:**

This work considers the supervised learning problem of fitting labels generated from a linear model with additive Gaussian noise $y_{i} = \theta^{\top}x_{i}+\varepsilon_{i}$ for three generative models for the features $x_{i}\in\mathbb{R}^{m}$: (a) *Isotropic model*: i.i.d. Gaussian features $x\sim\mathcal{N}(0,I_{m})$; (b) *Latent model*: noisy features generated from a latent Gaussian vector $x_{i} = Dz_{i}+e_{i}$ with $z_{i}\sim\mathcal{N}(0,\alpha I_{d})$; (c) *Latent model with orthogonal noise*: similar to the latent features model, but with the noise orthogonal to the features $x_{i} = Dz_{i}+D_{\perp}e_{i}$ where $D^{\top}D_{\perp}=0$. Given a training set $(x_{i},y_{i})$, $i=1,\cdots,n$, two scenarios are considered: the statistician either (1) fits the labels by first performing PCA on the features followed by linear regression on their projection on the top singular vectors; (2) performs PCA on a pre-training set $(x_{i})$ for $i=1,\cdots,n_{p}$ and performs linear regression on the projection of the training data on the top singular vectors learned in the pre-training phase. In either cases, the number of principal vectors might not coincide with the dimension of the data latent space. This is compared with performing linear regression directly on the features.

The two main theoretical results are:

- *Theorem 1*: an exact asymptotic characterisation of the generalisation error in terms of a bias-variance decomposition of model (1) in data model (a).
- *Theorem 2*: a concentration bound for the training risk of estimating the eigenvectors of the optimal decoder.

And the main conclusions of the work are:

1. To show that performing PCA before regression has an implicit regularisation effect, mitigating for instance the interpolation peak in this model. Despite this, doing PCA might also harm the risk in the regime where one has more features than data.

2. Pre-training the PCA features can lead to better estimation performance in the regime where one has more features than data. However, increasing the quantity of pre-training data only decreases the risk up to the point where the optimal decoder is perfectly learned.


**Summary Of The Review:**

Overall, I find the discussion in this work interesting. However, the fact that it is supported by incomplete theoretical results is deceiving, and hinders a finer understanding of the phenomenology reported. I think that this contribution could be of potential interest to the ICLR community if the theoretical part could be further developed.

---

> ### Author Response · Authors · 2022-11-14
> **Author response (2/2)**
>
> **References**:
>
> [Xu and Hsu, 2019] Ji Xu and Daniel J Hsu. On the number of variables to use in principal component regression. Advances in neural information processing systems, 32, 2019.
>
> [Mei & Montanari, 2022]Song Mei and Andrea Montanari. The generalization error of random features regression: Precise asymptotics and the double descent curve. Communications on Pure and Applied Mathematics, 75(4):667–766, 2022.
>
> [Gerace et al., 2020] Federica Gerace, Bruno Loureiro, Florent Krzakala, Marc Mezard, and Lenka Zdeborova. Generalisation error in learning with random features and the hidden manifold model. In International Conference on Machine Learning, pp. 3452–3462. PMLR, 2020.
>
> [Hastie et al., 2022] Trevor Hastie, Andrea Montanari, Saharon Rosset, and Ryan J Tibshirani. Surprises in highdimensional ridgeless least squares interpolation. The Annals of Statistics, 50(2):949–986, 2022.
>
> [Goldt et al., 2020] Sebastian Goldt, Marc Mezard, Florent Krzakala, and Lenka Zdeborova. Modeling the influence of data structure on learning in neural networks: The hidden manifold model. Phys. Rev. X, 10: 041044, 2020.

---

> ### Author Response · Authors · 2022-11-14
> **Author response (1/2)**
>
> We thank the reviewer for the highly detailed analysis of our paper and that you consider our work to have "many potential follow ups". We think your questions and comments helped us improve our submission. We address your points individually below.
>
> **Answer to weaknesses**:
> - We note that we use (as you also acknowledge) two data generators. *(a) isotropic* and *(b) latent variable data*. The orthogonal one (c) is only a special case of (b) for technical reasons. The PCA-regression model is the same throughout the paper but trained differently in Section 5 (what you call model (2)).
> - We present results on the asymptotic risk for (1)-(a), but (1)-(b) is an open problem.
> - For pre-training, our Theorem 2 provides the missing gap to make the more general asymptotic results of [Xu and Hsu, 2019] useful in practice. We work this out more clearly now in Section 5.2. Hence, the "informal discussion" is replaces by a formal connection to this prior work to highlight our contribution clearly.
> - Note: in your assessment of pre-training you write about "a concentration bound on the estimation of the decoder" which should be the "encoder" (the PCA).
>
> **Questions**:
> - **[Q1]**: In the original submission, the solid lines were the results of numerical simulation (and therefore noisy). We changed this throughout the paper: solid lines now show the analytical solution and $\times$-marks show numerical simulation results.
> - **[Q2]**: We analyzed this behavior in more detail and realized that this is most likely an artifact of our experimental setup. The additional experiments for larger ranges of $\gamma$ and different values of $\alpha$ are in Appendix E2.
> - **[Q3]**: Since the connection to Ridge regression is not one of the main contributions of the paper but rather an 'interesting connection', we moved it to Appendix E3. In Appendix E3.2 we discuss the finding that $\lambda\rightarrow\infty$ seems to be the best regularization strength in our setup and compare it with finding from previous studies. While [Mei & Montanari, 2022; Gerace et al., 2020] both work with nonlinear models, [Hastie et al., 2022] uses a linear model which is closer to our setup. We discuss the connection to prior work which found $\lambda=\infty$ as the best regularization in that context as well.
>
> **Comments**:
> - **[C1]**: Throughout the paper, we avoided the terms over- and underparameterization since the PCA-model is not itself overparameterized due to the low-dimensional embedding. We therefore corrected the phrasing above Figure 2 to low- and high-dimensional inputs.
> - **[C2]**: We have to correct the reviewer here, that $\mu>1$ already denotes the case "of interest" with more pre-training data as training data ($\mu=\frac{n_p}{n}$). Therefore, our analysis concentrates on the practicaly relevant part. $\mu<1$ is not practical as we could always use the $n$ training samples also for pre-training to obtain at least $\mu=1$.
> - **[C3]**: We thank the reviewer to point us to this prior work of similar data-generating models. We refer to this e.g. [Goldt et al., 2020] or [Gerace et al., 2020] for related data-generating processes such as the hidden manifold model and discuss the Gaussian equivalence in the related work section. The data-generating process that we study is different in that the Gaussian equivalence only holds if $d\rightarrow\infty$ which we do not assume in our work.
>
> **Smaller typos and suggestions**:
> - Thank you! We are grateful for your detailed reading of the submission to spot the typos. We have fixed them accordingly.
> - We clarified through the use of solid lines for analytical solutions and $\times$-marks for numerical solutions the plots. Further, we clarified in the figure descriptions what can be seen in each plot (simulation or analysis results).
> - We thank the reviewer for pointing us to prior work on double descent which we included in our related work.

---

### Official Review · Reviewer_7G3J · 2022-10-25

**Confidence:** 4
**Correctness:** 3
**Technical Novelty And Significance:** 2
**Empirical Novelty And Significance:** Not applicable
**Recommendation:** 3

**Clarity, Quality, Novelty And Reproducibility:**

**Clarity & quality:** the writing is mostly easy to follow. A few minor comments:

1. In Figure 2, it is better to use continuous curves for the analytical solutions, and crosses for the empirical values (since they fluctuate around the asymptotic values).
2. In the latent variable experiments, how is $\mathbf{\theta}$ created? As previously mentioned, the risk depends on the alignment between the true coefficients and the features, so I do not think it is sufficient to only specify the expected magnitude $\mathbb{E} (\mathbf{\theta}^\top\mathbf{z})^2$.

**Novelty:** see weakness above.

**Reproducibility:** N/A.

**Strength And Weaknesses:**

## Strength

The paper studies a relevant and fairly well-motivated problem: since dimension reduction in PCA can be interpreted as a form of "regularization", the least squares estimator on the truncated features might be able to avoid double descent. The main text is fairly easy to read, and the precise asymptotic results nicely illustrates the benefit of PCA close to the interpolation threshold.

## Weaknesses

My main concern is that the theoretical results are somewhat incremental and underwhelming.

1. The benefit of PCA in avoiding double descent has already been shown in [Teresa et al.]. The authors claim that the main improvement is the asymptotic result, but this is only shown for isotropic data, and the analysis follows from standard random matrix computation similar to that in [Hastie et al.]. Moreover, the observation that dimensionality reduction can suppress the peak by improving the stability of the pseudo-inverse is rather intuitive.

2. While [Teresa et al.] does not provide the asymptotic risk formula, it highlights the role of alignment between the true coefficients and the features. This alignment is known to impact the performance of both ridge regression and PCR as shown in [Wu and Xu], but cannot be captured by the isotropic data model. In this submission the authors also considered a latent variable model with (anisotropic) decaying eigenvalues, but the analytical solution of the risk is not derived.

3. The authors should also discuss the difference in the observed phenomena between the studied PCA-regression model and the PCR model in [Xu and Hsu] that assumes access to the population covariance, which can be obtained in an unsupervised manner. The precise risk of the population PCR estimator has been analyzed in [Wu and Xu] for general features and true coefficients, and conditions under which low-dimensional projection does not benefit the model performance are also given.

4. The comparison between the PCA model and ridge regression is not quantitative. Note that the asymptotic risk of both estimators are available in the isotropic setting. Hence it would be nice to quantitatively compare the performance of the two models (e.g. under optimal truncation and regularization for a given SNR).

5. [minor] The data-generating process that assumes a low-dimensional structure is sometimes termed the "hidden manifold model", for which the precise risk of two-layer networks has been studied in many prior works, see [Gerace et al.].


Xu and Hsu 2019. On the number of variables to use in principal component regression.
Hastie et al. 2019. Surprises in high dimensional ridgeless least squares interpolation.
Teresa et al. 2020. Dimensionality reduction, regularization, and generalization in overparameterized regressions.
Wu and Xu 2020. On the optimal weighted $\ell_2$ regularization in overparameterized linear regression.
Gerace et al. 2020. Generalisation error in learning with random features and the hidden manifold model.


**Summary Of The Paper:**

The paper studies the risk (generalization error) of the PCA least squares estimator and shows that dimension reduction can avoid the peaking in the risk curve. The analysis is divided into two parts: (i) precise bias-variance decomposition in the proportional limit for isotropic data, and (ii) non-asymptotic bound on the error due to estimating the low-dimensional projection in an unsupervised manner. Theoretical results are supported by some small-scale experiments.

**Summary Of The Review:**

In my opinion this submission is below the acceptance bar due to the incremental theoretical contribution. I am happy to update my evaluation if the authors can adequately discuss the prior results and clarify the novelty in their theoretical analysis.

---

> ### Author Response · Authors · 2022-11-14
> **Author response (1/2)**
>
> We thank the reviewer for the very thoughtful assessment of our work. We believe that the paper update due to your review with the connection to previous work improved the overall point of the paper. Let us address the reviewer's concerns pointwise.
>
> **Answers to weaknesses**:
> 1. We agree that [Huang et al., 2022] (=[Teresa et al., 2022] since the bibtex name entry was wrong; middle name instead of last name) does indeed show that the double descent is avoided with PCA-regression. We also agree that this behavior is not surprising due to the regularizing behavior of the PCA as we state in Section 6. [Huang et al., 2022] further provide upper and lower bounds on the non-asymptotic risk. For them to be tight, they rely on the alignment between the population covariance and the estimated covariance by using the same result from [Loukas, 2017] as we do for the concentration bound in Theorem 2. Novel in our work are 1) the asymptotics for isotropic data which provide a clear connection to the results for the direct regression model in [Hastie et al., 2022] as a special case; and 2) the numerical results on the latent data generator which provide insights into the PCA-regression model on realistic data structures. We hope that the connection in 1) together with the results from 2) can be exploited to derive asymptotics for more general covariance structures such as our latent variable data generator in future works.
> 2. The alignment between population covariance and estimated covariance which impacts the risk as [Wu and Xu, 2020] show is indeed exploited in [Huang et al, 2022]. In the latter work, the authors make use of the results in [Loukas, 2017] to derive their bounds. In a similar way, we use [Loukas, 2017] to derive the sample complexity in Theorem 2 which connects these results and therefore fills an important gap as we argue in our answer to point 3 below.
> 3. [Xu and Hsu, 2019] study principle component regression (PCR) which is the same our PCA-regression model. The authors obtain asymptotic results for more general structures of the covariance matrix. However, their main assumption is to have access to the true covariance matrix. [Wu and Xu, 2020] extend [Xu and Hsu, 2019] to consider the (mis-)alignment under the same assumption of access to the true covariance. [Xu and Hsu, 2019, section 4] state that the true covariance can be "estimated [...] very accurately via unlabeled data". We provide this connection with our Theorem 2 by introducing a sample complexity for when this estimation is sufficiently correct for PCA-regression. We updated our paper in section 5.2 to work this out clearly. We updated the contribution accordingly to highlight this connection more clearly.
> 4. Since the connection to Ridge regression is not the main focus of the paper, we moved this section to Appendix E3. We included a quantitative comparison with known asymptotic results from [Hastie et al., 2022] for different strength of Ridge regularization $\lambda$ and for the optimal value $\lambda^\ast$, which depends on $\gamma$. A discussion about this comparison is provided. Due to the nature of the isotropic data, the two models show qualitative different behavior here. A closer relationship is notable for latent variable data as we noted in the original version of the paper (now Appendix E3.2).
> 5. [minor] We are thankful for pointing us to this line of prior work. We included references e.g. [Goldt et al., 2020] or [Gerace et al., 2020] for related data-generating processes such as the hidden manifold model throughout the paper and discussed it specifically in the related work.
>
> **Answers to clarity & quality**:
> 1. We are happy for this suggestion and followed it throughout the updated submission. We now use $\times$-marks for all numerical simulation results and solid lines only for the analytical solutions. We further clarified the origin of the plotted data as `simulation` or `analysis` to distinguish it more clearly.
> 2. We defined this in Appendix C, (33) but made this point clear now.
>
> We are hopeful to have addressed the points of the reviewer about the connection to prior work. Discussing this more clearly made the contribution of the paper stronger in our mind as we fill a crucial step to use the asymptotic results from [Xu and Hsu, 2019] in practice.

---

> ### Author Response · Authors · 2022-11-14
> **Author response (2/2)**
>
> **References**
>
> [Huang et al., 2022] Ningyuan (T.) Huang, David W Hogg, and Soledad Villar. Dimensionality reduction, regularization, and generalization in overparameterized regressions. SIAM Journal on Mathematics of Data Science, 4(1):126–152, 2022.
>
> [Hastie et al., 2022] Trevor Hastie, Andrea Montanari, Saharon Rosset, and Ryan J Tibshirani. Surprises in highdimensional ridgeless least squares interpolation. The Annals of Statistics, 50(2):949–986, 2022.
>
> [Loukas, 2017] Andreas Loukas. How close are the eigenvectors of the sample and actual covariance matrices? In International Conference on Machine Learning, pp. 2228–2237. PMLR, 2017.
>
> [Wu and Xu, 2020] Denny Wu and Ji Xu. On the optimal weighted ℓ2 regularization in overparameterized linear regression. Advances in Neural Information Processing Systems, 33:10112–10123, 2020.
>
> [Xu and Hsu, 2019] Ji Xu and Daniel J Hsu. On the number of variables to use in principal component regression. Advances in neural information processing systems, 32, 2019.
>
> [Goldt et al., 2020] Sebastian Goldt, Marc Mezard, Florent Krzakala, and Lenka Zdeborova. Modeling the influence of data structure on learning in neural networks: The hidden manifold model. Phys. Rev. X, 10: 041044, 2020.
>
> [Gerace et al., 2020] Federica Gerace, Bruno Loureiro, Florent Krzakala, Marc Mezard, and Lenka Zdeborova. Generalisation error in learning with random features and the hidden manifold model. In International Conference on Machine Learning, pp. 3452–3462. PMLR, 2020.

---

### Official Review · Reviewer_YsaP · 2022-10-27

**Confidence:** 4
**Correctness:** 3
**Technical Novelty And Significance:** 3
**Empirical Novelty And Significance:** Not applicable
**Recommendation:** 6

**Clarity, Quality, Novelty And Reproducibility:**

Clarity: In general quite clear, although there are small things that can be improved, e.g., define \gamma when first mentioning it, tell us what is the value of d for Figure 2 (experiment with isotropic features).

Quality: Overall good. Analytical solution matches well with numerical results (partially thank to the M-P law). I have a question on Theorem 2 -- can the author comment on whether the O(1/np) dependency is tight or not?

Reproducibility: My best guess is that the results can be reproduced.

Novelty: I am not in a good position to judge the novelty with high confidence given that there are a lot of work related to double-descent in recent few years and I have not been able to track them. The results in the paper seem relatively straightforward (which does not mean it is not novel) and slightly fragmented, and overall I'm on the weakly positive side.

**Strength And Weaknesses:**

Strength:
The paper is relatively clear and considers a relatively simple setting for theoretical understanding. Within a simple setting, it reveals that if the implicit dimension is fixed, there is no double descent behavior as input dimension and sample size change. The 'limitations' paragraph in section 6 point out the weaknesses of the current work.

Other weakness:
The analysis is based on a simplified setting (not a big deal), and the theorem from each section is based on a set of its own simplified assumptions. For example, assumption (15) is clearly for analytical purposes and I don't think it is necessary.

**Summary Of The Paper:**

The paper studies the setting where response y and input x follow a linear transformation of the latent variable z, and the goal is to learn y from x through PCA regression. The paper shows that for fixed PCA dimension, there is no double-descent phenomena as the ratio between dim(x) and sample size n varies (the theorem is for isotropic data). Numerical experiments in multiple data settings verify the theoretical finding. The paper shift to discuss a pre-training setting where more data is allowed for the PCA step and built theoretical analysis over a slightly different data assumption.

**Summary Of The Review:**

The paper shows some interesting results on the asymptotic phenomena of supervised learning with high-dim inputs but reduced to low(and fixed)-dim space through dimension reduction. The analysis are based on somewhat simplified settings, but are consistent with numerical observations. Without judging too much on the significance and novelty of the paper, I'd put my rating as "marginally above the acceptance threshold". If more experience reviewer challenges the novelty/significance of the paper, my rating would as a result be less confident.

---

> ### Author Response · Authors · 2022-11-14
> **Author response**
>
> We thank the reviewer for their analysis of our submission. We are happy to read that our paper is "in general quite clear" and we are positive about your recommendation. Your summary of our submission is highly accurate. In our opinion, you have assessed the strength and weaknesses of the submission precisely.
>
> Let us address your points of concern individually:
> - The concentration bound in Theorem 2 is the tightest to the best of our knowledge. [Huang et al., 2022] base their results similar to ours on [Loukas, 2017] to improve their bounds (which are non-asymptotic compared to ours). However, there is no guarantee that this is the tightest bounds and it is possible that improved bounds with better convergence rates could be derived.
> - We included the dimension of $d$ for Figure 2. In isotropic data models $d=m$.
> - We note that $\gamma$ was already defined at the end of the introduction (first bullet point) and in Theorem 1 again.
>
> We would like to highlight two important additions in the paper update:
> - we included results from real-world data sets from genetics to confirm our numerical observations, see Figure 4. We believe that this improves the submission substantially.
> - we highlight that Theorem 2 provides the missing link to asymptotic results from [Xu and Hsu, 2019] which makes their results now useable in practice.
>
> **References**:
>
> [Huang et al., 2022] Ningyuan (T.) Huang, David W Hogg, and Soledad Villar. Dimensionality reduction, regularization, and generalization in overparameterized regressions. SIAM Journal on Mathematics of Data Science, 4(1):126–152, 2022.
>
> [Loukas, 2017] Andreas Loukas. How close are the eigenvectors of the sample and actual covariance matrices? In International Conference on Machine Learning, pp. 2228–2237. PMLR, 2017.
>
> [Xu and Hsu, 2019] Ji Xu and Daniel J Hsu. On the number of variables to use in principal component regression. Advances in neural information processing systems, 32, 2019.

---

### Author Response · Authors · 2022-11-14
**General answer / summary of changes**

We thank all reviewers for their thoughtful and in-depth reviews. It is promising to see that you consider the problem that we study is of "potential interest to the ICLR community" [JqQv], "relevant and fairly well motivated" [7G3J] and that our writing is clear [YsaP, eejS] and easy to read [7G3J].

Your suggestions and questions helped to update the submissions in several parts. Therefore, we would like to highlight the changes of the new update to the original submission. We:
- included a real-world example with a genetics data set in Section 4.2 and Figure 4. We observe key features from our simulations in this example.
- re-wrote the former high-level discussion about the risk for pre-training in Section 5.2 (paragraph `Connection to risk`). We now make it clear that our theoretic results in Theorem 2 fill the gap when the asymptotic results from [Xu and Hsu, 2019] can be used in practice. We updated the contribution list and conclusion to mention this more explicitly.
- updated related work to highlight our novelty compared to previous work on generalization for PCA-regression models. We also added two sections to the related work (one is placed in Appendix A due to space limitations) to put the submission into more related context: 1) the relation of our latent variable data generator to previous data models such as the hidden manifold model and 2) the use of PCA-regression in real-world applications.
- changed all plots such that analytical solutions are shown by solid lines and numeric simulations by $\times$-marks. Additionally, we highlight `simulation` or `analysis` in the caption to be more clear about what results are visualized.
- moved the connection of our PCA-regression model with Ridge regression to Appendix E3 as it is not the focus of the work. We also included a comparison with analytical solution in the isotropic case (from [Hastie et al., 2022] in Appendix E3.1
- discussed the phenomenon that the supervised risk for $\alpha>0$ in Figure 3 is slightly higher for PCA-regression than for direct regression in $\gamma>1$ and included additional experiments (varying $\alpha$; larger $\gamma$). We note that this phenomenon is mainly a result of our experimental setup and within the uncertainty.
- added experiments for the pre-training case in Appendix G. We show here that concentrating on the well-specified case $\hat d=d$ in the main paper is justified as the results from Section 4 (fully supervised) about model misspecification translate to this setting.
- incorporated all minor feedback such as missing transposes $\top$ and changes in phrasing.


**References**

[Hastie et al., 2022] Trevor Hastie, Andrea Montanari, Saharon Rosset, and Ryan J Tibshirani. Surprises in highdimensional ridgeless least squares interpolation. The Annals of Statistics, 50(2):949–986, 2022.

[Xu and Hsu, 2019] Ji Xu and Daniel J Hsu. On the number of variables to use in principal component regression. Advances in neural information processing systems, 32, 2019.

---

### Decision · Program_Chairs · 2023-01-20

**Decision:**

Reject

**Justification For Why Not Higher Score:**

The models considered are fairly simple (linear regression), so the weight on the paper falls on the quality of the theoretical results. Given that the only "full" result concerns a setting that's fairly well studied, the paper may be better suited to a more specialized, statistics-oriented conference (e.g., AISTATS).

**Justification For Why Not Lower Score:**

N/A

**Metareview: Summary, Strengths And Weaknesses:**

The paper concerns (two) linear regression models: on in which data is isotropic, the other in which there is planted, latent, linear structure.  The authors prove there is no double-descent behavior of (two variants) of PCA + linear regression on such data models. The authors provide a combination of theoretical results and simulations for these combinations of models and algorithms. The reviewers appreciate some of the technical contributions of the paper. However, the theoretical results are really "fully complete" only for the isotropic case (which has been studied rather extensively in several prior works, see the discussion with reviewer 7G3J for a complete list), and characterizing the asymptotic risk for the "planted" case is an "open problem" (in the words of the authors), though some partial, suggestive results are shown in the manuscript.